# Magnetically tightened form-stable phase change materials with modular assembly and geometric conformality features

Yongyu Lu[1,2,4], Dehai Yu[1,4], Haoxuan Dong[1], Jinran Lv[1], Lichen Wang[2], He Zhou[3], Zhen Li[1], Jing Liu [2✉] & Zhizhu He [1✉]

Phase change materials have attracted significant attention due to their promising applications in many fields like solar energy and chip cooling. However, they suffer leakage during the phase transition process and have relatively low thermal conductivity. Here, through introducing hard magnetic particles, we synthesize a kind of magnetically tightened form-stable phase change materials. They achieve multifunctions such as leakage-proof, dynamic assembly, and morphological reconfiguration, presenting superior high thermal (increasing of 1400–1600%) and electrical (>$10^4$ S/m) conductivity, and prominent compressive strength, respectively. Furthermore, free-standing temperature control and high-performance thermal and electric conversion systems based on these materials are developed. This work suggests an efficient way toward exploiting a smart phase change material for thermal management of electronics and low-grade waste heat utilization.

[1] Department of Vehicle Engineering, College of Engineering, China Agricultural University, Beijing 100083, China. [2] Key Laboratory of Cryogenics and Beijing Key Laboratory of Cyro-Biomedical Engineering, Technical Institute of Physics and Chemistry, Chinese Academy of Sciences, Beijing 100190, China. [3] School of Materials Science and Engineering, University of Science and Technology Beijing, Beijing 100083, China. [4]These authors contributed equally: Yongyu Lu, Dehai Yu. ✉email: jliu@mail.ipc.ac.cn; zzhe@cau.edu.cn

Phase change materials (PCMs) are such a class of materials that absorb or release large amounts of heat while their temperature keeps constant during the melting or solidifying process. In thermal energy storage (TES) systems, latent heat storage has distinct advantages over sensible heat storage or thermochemical reactions due to its high energy density with a slight temperature swing[1–4]. PCMs play a vital role in the latent heat storage technique, and is regarded as a promising technology to cope with energy and environmental crisis and thus has been extensively studied in recent years[5–8]. It was demonstrated that PCMs own big potential in those areas like solar energy, load shifting/leveling, and waste heat recovery[7,9–12]. They would help alleviate the discontinuous and intermittent nature of solar irradiation and offset the mismatch between electricity supply and demand[13–15]. In addition, practical applications of PCMs are constantly widened ranging from electronic cooling, green buildings, smart textiles to infrared stealth with the deepening of research[16–18].

Despite the tremendous progress that has been made in developing PCMs, their performance still contains several shortcomings, such as low thermal conductivity, liquid leakage, supercooling, and phase separation[6,18]. Although supercooling and phase separation behaviors deteriorate the storage capacity and thermal stability of PCMs, these two behaviors only occur in hydrate salts[19]. The common and critical bottlenecks of PCMs are their low thermal conductivity and leakage issue[5,20–23]. Inherent low thermal conductivity limits the heat transfer within PCMs and consequently leads to the reduction of thermal diffusion rate. The main way to tackle this problem is incorporating metal-, carbon- or ceramic-based high-thermal-conductivity additives into PCMs matrix[1,5,8,20]. Compared with the issue of thermal conductivity, liquid leakage appears as a more troublesome agenda that is worth concentrating on, as fluid PCMs can lead to contamination, corrosion, or even short circuit of equipment. It is for such reason, considerable works have been dedicated to resolving this issue, such as encapsulating PCMs with shells to produce core-shell-like capsules[11,13,24,25]. However, the shell usually presents a weak thermal response and influences the thermophysical properties of PCMs, and there exists the risk of rupture under large thermal stress[26]. More importantly, though adding high thermal conductivity materials and PCMs encapsulation can compensate for the abovementioned defects of PCMs, introducing excessive additives sacrifices their energy storage density. In this case, employing three-dimensional (3D) structural substances with high thermal conductivity to make form-stable PCMs becomes the focus of research[5,17,27]. The 3D structural substances act as not only the thermal conductivity promoter but also supporting materials, which significantly decreases the filler loading of PCMs[17]. Among these substances, 3D carbon materials and metal foams are the most frequently used additives[28,29]. Carbon materials generally exhibit low dimensional morphology. It is particularly difficult for them to construct an interconnected thermal conduction framework, and the resulting materials often present low thermal conductivity due to the loose contact, which means the increase in thermal conductivity of the as-prepared form-stable PCMs is rather limited[30,31]. As for metal foams, it is not easy to impregnate PCMs into their inner pores and obtain thoroughly filled composites because of the large surface tension or insufficient wetting of PCMs to the metal[32]. Up to now, the effective method to synthesize form-stable PCMs is still lacking and further explorations are in demand[33].

In this work, we introduce hard magnetic particles as the supporting material into PCMs. Applying an external magnetic field, these hard magnetic particles agglomerate together forming a 3D cluster with abundant interparticle pores and such structure can keep intact after removing the magnetic field[34]. Under the capillary action, liquid PCMs fill inner micropores forming liquid bridges and then being stabilized inside the magnetic cluster, which enables PCMs to be leakage-proof. To synthesize this form-stable PCM, here as a proof of concept, we select NdFeB and paraffin, the most commonly used hard magnetic material and organic PCM, as the input for our system. NdFeB particles are surface modified to NdFeB@Ag through in situ silver-plating to improve their thermal conductivity. Different from conventional supporting materials which must possess or build a prerequisite 3D structure, NdFeB@Ag particles can directly mix with paraffin like general particles. In the following magnetization process, they align along the magnetic field constructing an oriented network thereby fabricating a magnetically tightened form-stable phase change material (MTPCM), as illustrated in Fig. 1a. Our approach ensures saturated filling of PCMs between tightly connected NdFeB@Ag particles, eliminating the universal incomplete filling and incompact contact problems in form-stable PCMs. It needs to be emphasized that the 3D structure made of magnetic-oriented hard magnetic particles is the core of MTPCMs. It endows MTPCMs with a robust structure and magnetism. As a result, they not only demonstrate the leakage-proof ability and shape stability during the phase change process but also exhibit magnetic assembly and shape reconfigurable features. In contrast to pristine paraffin, the interconnected structure inside MTPCMs constitutes effective heat transfer pathways, coupling with high-thermal-conductivity NdFeB@Ag particles. This synergistically speeds up the thermal charging/discharging rate, leading to 14~16 times enhancement in thermal conductivity for MTPCMs. In addition, MTPCMs display excellent compressive strength that is also attributed to their supporting architecture. In the end, we explore the application of MTPCMs in energy conversion and thermal management aspects, and the results reveal that MTPCMs own superior energy conversion efficiency and temperature control capacity.

## Results

**MTPCMs and their features**. As a basic unit in the 3D structure, the thermal conductivity of NdFeB particles will directly impact the heat transfer of MTPCMs. Considering this, we adopted in situ chemical plating method to coat a silver shell on the surface of micro NdFeB particles modifying them to NdFeB@Ag particles (details see Supplementary Fig. 1 and the "Method" section)[35]. As shown in Fig. 1b, the color of magnetic particles changed from dark black to yellow which was similar to the silver microparticles, indicating NdFeB particles are completely covered by silver shells. This was also demonstrated by scanning electron microscopy (SEM) images and the corresponding energy-dispersive X-ray spectroscopy (EDX) element mappings in Supplementary Fig. 2. In contrast to bare NdFeB, the core particle was wrapped in a continuous silver layer, and there existed only the silver element on the surface of NdFeB@Ag. The homogeneous composite of NdFeB@Ag particles and paraffin can be obtained by simply adding and stirring NdFeB@Ag particles into melted paraffin. Correspondingly the resulting composite presented a yellow appearance (Fig. 1c). Applying a magnetic field to the composite, NdFeB@Ag particles aggregated and formed a porous structure that provided the strong capillary force to stabilize paraffin. In this way, the magnetically tightened form-stable phase change material (MTPCM) was fabricated. The mixing ratio of NdFeB@Ag particles is an important parameter for MTPCMs as they compromise their energy storage density for the leakage-proof and form-stable abilities. It was explored that the lowest volume mixing ratio for MTPCMs was 15.84%, and as a comparison, we prepared three samples with different volume ratios of 15.84%, 19.12%, and 23.31% in this paper. Unless

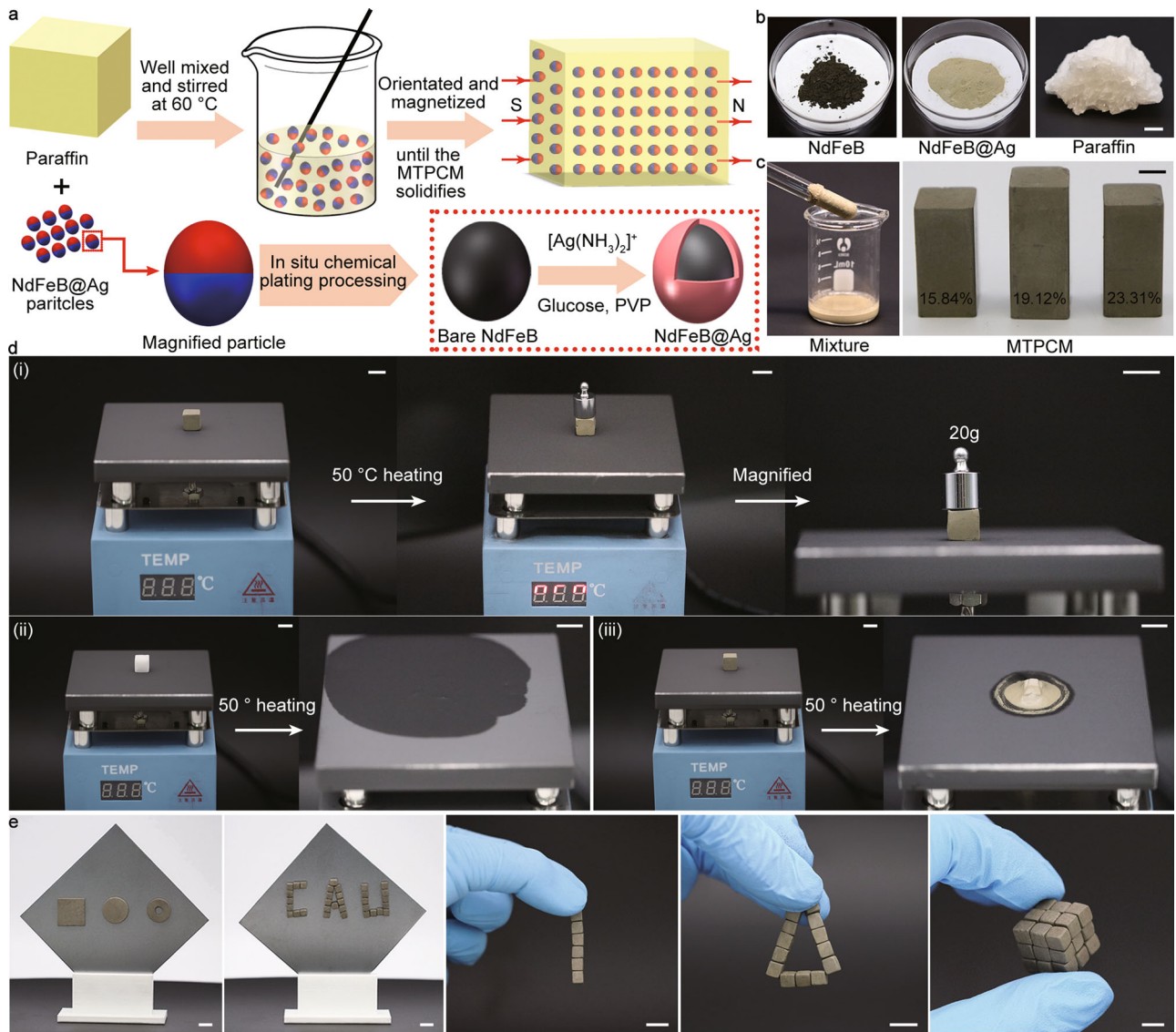

**Fig. 1 Magnetically tightened form-stable phase change materials (MTPCMs) with magnetic assembly and conformal shape features. a** Schematic procedure for fabricating MTPCMs. **b** Photographs of NdFeB, NdFeB@Ag, and paraffin. Scale bar: 5 mm **c** The mixture of NdFeB@Ag particles and paraffin, and MTPCMs with different volume ratios. Scale bar: 5 mm. **d** Shape stability and leakage behaviors of (i) MTPCM (ii) paraffin and (iii) unmagnetized composite during the phase change process. Scale bars: 10 mm. **e** Modular assembly and conformal geometry features of MTPCMs. Scale bars: 10 mm.

specified, the composite refers to the unmagnetized mixture of NdFeB@Ag particles and paraffin in the following context to facilitate elucidation.

It is deduced that MTPCMs have leakage-proof and shape stability behaviors based on the theoretical analysis. To verify this point, a leakage test was performed on the heating platform at 50 °C for MTPCM, unmagnetized composite, and pristine paraffin, which were all processed into $10 \times 10 \times 10$ mm cubes. As shown in Fig. 1d, after being heated for a while, pristine paraffine melted completely into a fluid spreading out on the plate. As for the unmagnetized composite, phase separation occurred during the phase change process, paraffin flowed away from the composite leaving a stack of collapsed NdFeB@Ag particles. Unlike the above two materials, no leakage was observed on the MTPCM block, and it always retained its initial shape even after being compressed by a weight of 20 g, which is ascribed to the magnetically-induced strong capillary force inside the material. The test result confirms the leakage-proof and shape stability capacity of MTPCMs. This conclusion was presented

more vividly by hanging the three samples and heating them via a heat gun. During the phase change process, the paraffin and unmagnetized composite melted and dropped on the table while the MTPCM keeps always hanging on the string with its original shape (see Supplementary Movie 1). Furthermore, we took the 15.84% MTPCM as a sample and heated it at 50 °C for over two hours in the oven. It can be seen that the weight of this sample did not change at all before and after heating, demonstrating the excellent leakage-proof ability of MTPCMs again (see Supplementary Fig. 3).

Hard magnetic particles can reassemble and then rebuild an interconnected framework when their original structure is destroyed by external stimuli, which enables MTPCMs a shape transformable function. Only by a simple heating operation, the MTPCM transforms from cylinder to cuboid while maintaining its leakage-proof ability intact (Supplementary Movie 2). This function makes it convenient to create or reconfigure MTPCMs to diverse geometries according to the specific application. Moreover, we fabricated MTPCMs into standard modules to

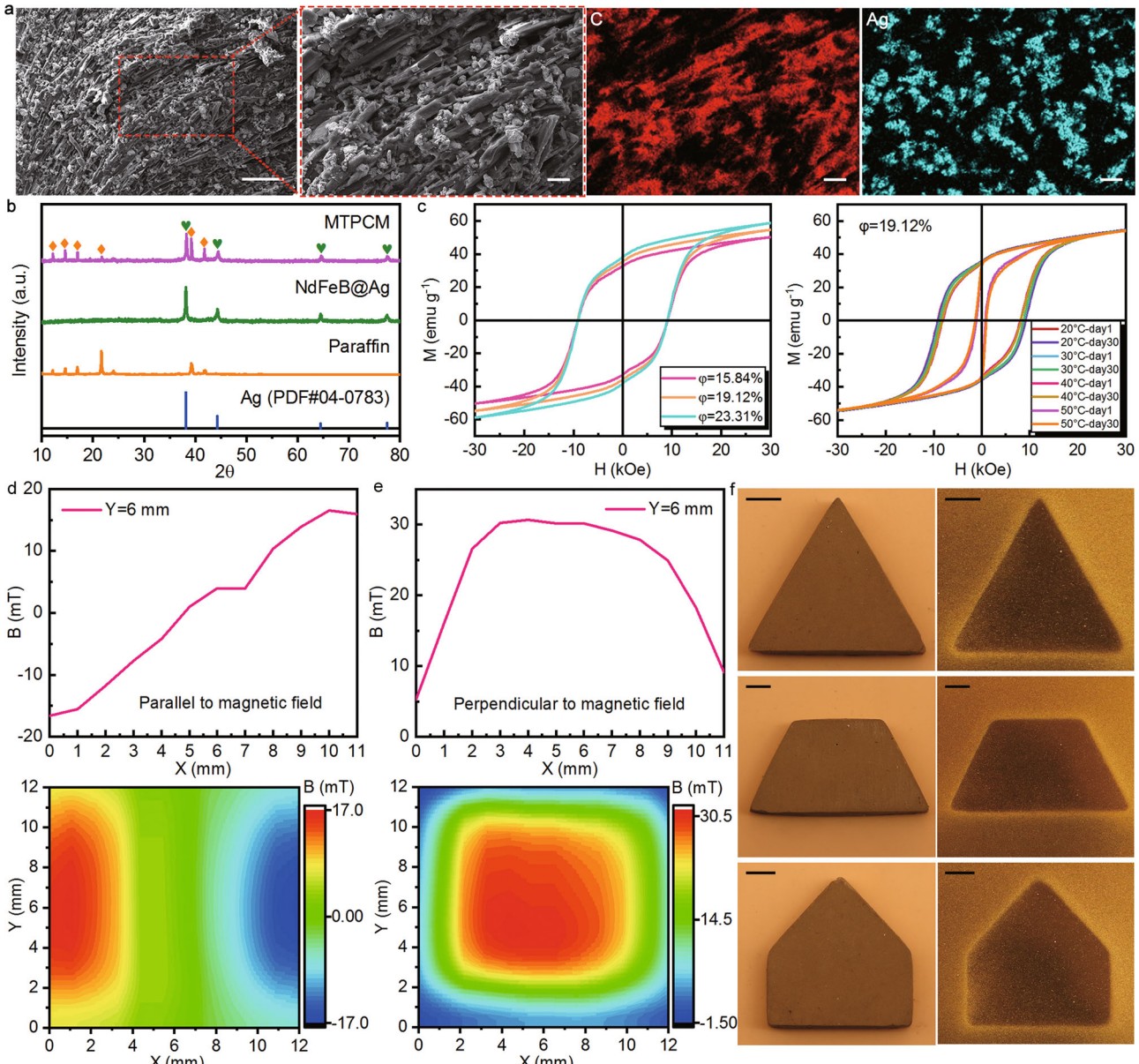

**Fig. 2 Characterizations of magnetically tightened form-stable phase change materials (MTPCMs). a** SEM images and the corresponding EDX element mappings for the cross-section of MTPCMs, revealing the directional alignment of NdFeB@Ag particles induced by the magnetic field. Scale bars: 50 μm, 10 μm, 10 μm and 10 μm. **b** Comparison of XRD patterns among MTPCM, NdFeB@Ag, paraffin, and the PDF card of pure Ag. **c** Magnetic hysteresis loops of MTPCMs with different volume ratios and the typical MTPCM sample at different temperatures at different time points. **d, e** Surface magnetic flux density of MTPCM in the direction along and perpendicular to the magnetic field. **f** Surface magnetic field distributions of MTPCMs with different shapes. Scale bars: 4 mm.

make better use of their magnetism. The modules can attach spontaneously with good contact like "magnetic lego" due to the magnetic attraction. To further extend this concept, a series of intricate 3D architectures with different shapes and structures were realized via the modular assembly, and the generating object displayed a robust structure (Fig. 1e and Supplementary Movie 3). The above conformal shape and modular assembly features render MTPCMs favorable as they make instantly customizing the shape of PCMs to meet complex practical requirements a reality.

The microstructure of the transverse section for MTPCMs was investigated using SEM and EDX. The cross-section images in Fig. 2a presented an oriented structure that is derived from the directional alignment of NdFeB@Ag particles and the

accompanying passively arrayed paraffin. The anisotropic tendency in structure inevitably brings anisotropy performance to some extent. Besides, there were almost no voids inside the MTPCM, which is beneficial to guarantee its thermal and mechanical performance. The X-ray diffraction (XRD) pattern of the MTPCM in Fig. 2b only contained peaks of paraffin and pure Ag since the NdFeB particles are completely covered by silver shells. The XRD test states that NdFeB@Ag particles and paraffin is a physical combination and there is no new substance formed, revealing the force to stabilize paraffin is merely from magnetic-induced capillary action. To explore the magnetic properties of MTPCMs, we measured their magnetic hysteresis loops and surface magnetic flux density, as displayed in Fig. 2c, d. With the increase of volume ratio for MTPCM samples, their saturation

magnetization and remanence rose from 50.22 to 58.72 emu g$^{-1}$ and 32.81 to 38.15 emu g$^{-1}$, illustrating the MTPCMs have a good magnetic response and strong magnetism. Furthermore, a typical sample with the 19.12% volume ratio was tested at different temperatures at different time points. The magnetic hysteresis loops tested at 20, 30, and 40 °C on day 1 and day 30 almost overlapped. Even though the magnetic loops of the sample had a smaller enclosed area at 50 °C, the values of saturation magnetization and remanence were nearly the same as those at other temperatures, which were about 54 and 35 emu g$^{-1}$. It is attributed to the high Curie temperature of NdFeB (315 °C) that protect the magnetism of MTPCMs from high temperature. The results prove that MTPCMs own stable magnetism, which will not be affected by heating or long-time storage. The strong magnetism correspondingly generates high surface magnetic flux density in MTPCMs. For a $10 \times 10 \times 10$ mm cubic MTPCM sample, the magnetic flux density on the surfaces which are parallel and perpendicular to the magnetic field reached up to 16.58 and 31.01 mT, respectively. Except for that, Fig. 2f shows the surface magnetic field profiles of MTPCMs with different shapes by magnetic observation cards. The samples all appeared a uniform magnetic field distribution and unambiguous boundary with their surroundings. The series of magnetic tests disclose that MTPCMs have not only strong and stable magnetism but also homogeneous magnetic distribution without any defects.

**Thermal and mechanical performance of MTPCMs.** The phase change properties of MTPCMs were characterized by differential scanning calorimetry (DSC) (Fig. 3a). The curves of pristine paraffin and MTPCM samples with different volume ratios were very close, and the thermal parameters were extracted and summarized in Fig. 3b. The onset melting and solidifying point of paraffin were 38.65 and 36.23 °C, respectively. The three MTPCM samples had nearly the same values with paraffin (38.95 and 37.33 °C for 15.84%, 39.5 and 37.13 °C for 19.12% and 40.05 and 37.07 °C for 23.31% sample). The fusion enthalpy for neat paraffin was 163.75 J cm$^{-3}$. With the addition of NdFeB@Ag particles, the value did not decrease but instead changed to 178.70, 172.59, and 141.63 J cm$^{-3}$ respectively, which was ascribed to that the magnetic particles increased the density of MTPCMs and thus improved their energy storage density. Moreover, unmagnetized composites were also measured by DSC and it was found that they had little difference with MTPCMs under the same volume ratio (Supplementary Fig. 4). These results indicate that paraffin plays the dominant energy storage role in MTPCMs. The NdFeB@Ag particles only serve as the supporting material, and their magnetism does not affect the phase change characteristics of paraffin. Figure 3c depicts the thermal gravimetric analysis (TGA) curves of paraffin and MTPCMs. Based on the data analysis, it can be achieved that the 5% weight loss temperature of MTPCM was higher than that of paraffin, and this temperature boosted with the increase of volume ratio in MTPCMs. On the contrary, the weight loss rate declined with the addition of NdFeB@Ag particles. It is speculated that the capillary pores inside MTPCM protect paraffin from evaporation to some degree and finally delay its decomposition. Hence, MTPCMs have better heat resistance and thermal stability than pristine paraffin. The heat charging/discharging rate and temperature-holding time of PCMs are significant issues concerning their applications and they can be assessed via the heating and freezing process[26]. The experimental system and temperature program are shown in Supplementary Fig. 5 and the Method section, and the MTPCM sample was produced into a $20$ mm $\times 20$ mm $\times 20$ mm cubic block. The cycle was repeated 10 times and the curves are recorded in Fig. 3d. In a magnified period, it can be observed that

the heat charging/discharging rate and temperature-hold time presented an inverse variation trend with the increase of volume ratio in MTPCMs. Specifically speaking, the sample with a larger volume ratio costs a shorter time to reach the equilibrium state in both heating and freezing processes, implying that it has a higher heat charging/discharging rate. And this results from the more NdFeB@Ag particles and thus builds better heat transfer pathways, accelerating the thermal storage and release efficiency of the sample. However, adding more NdFeB@Ag particles leads to the reduction of paraffin, and this will cut down the latent heat of the sample. Correspondingly, its phase change time, namely temperature-holding time, decreases. Hence, it needs to balance these two aspects of MTPCMs in practical applications. Thermal reliability is a fatal parameter for PCMs as it can evaluate their service life in practical applications. MTPCMs exhibited almost the same heat charging/discharging rate and phase change time in the above 10 cycles, presenting their potential for cycle reliability. To verify this point, over 1000 thermal cycles were repeated on a 23.31% sample (specific temperature setting could be seen in the "Methods" section). The temperature variation versus time of the sample was recorded in Fig. 3e. It could be discovered that there is no obvious fluctuation on the curve. To observe the cycle condition more clearly, we extracted the first, 102nd, 505th, and 1004th four cycles and plotted them in Fig. 3f. The trend of these four curves almost overlaps, proving the very prominent thermal reliability and durability of MTPCMs. As an essential property of PCMs, the thermal conductivity of MTPCMs was studied and depicted in Fig. 3g. The interconnected 3D structure of MTPCM offers efficient thermal conduction routes inside the entity. As the elementary unit on this network, modified NdFeB@Ag particles possess a silver-like high thermal conductivity. Under the synergistic effect of the above two factors, the thermal conductivity of MTPCMs is greatly enhanced by one order of magnitude compared to organic paraffin (0.21 W m$^{-1}$ K$^{-1}$), reaching 2.97, 3.11, and 3.41 W m$^{-1}$ K$^{-1}$ for three different volume ratio samples. As stated above, the MTPCMs exhibited an anisotropic structure. Given the heat transfer mechanism in MTPCMs, we investigated the thermal conductivity of MTPCMs in the direction of parallel and perpendicular to the magnetic field (Fig. 3h). It was easy to find that MTPCMs displayed higher thermal conductivity along the magnetic field direction and the difference of thermal conductivity in these two directions slightly rose with the increase of volume ratio in the samples (0.61 W m$^{-1}$ K$^{-1}$ for 15.84%, 0.7 W m$^{-1}$ K$^{-1}$ for 19.12% and 0.76 W m$^{-1}$ K$^{-1}$ for 23.31% sample). The anisotropic thermal performance in MTPCMs is mainly caused by the anisotropy in the structure, and this effect is reinforced with the increase of volume ratio in MTPCMs.

Applying a magnetic field, both soft and hard magnetic particles will rotate and align eventually forming chains along with the magnetic flux profiles[36,37]. The different point is soft magnetic particles collapse back into a pile of powders while hard magnetic particles can hold their configuration after removing the magnetic field. This characteristic arouses our inspiration to explore whether the alignment of NdFeB@Ag particles throughout the MTPCM can support the composite and strengthen its mechanical performance. On this background, the compression tests were conducted on the $10$ mm $\times 10$ mm $\times 10$ mm cubic MTPCM modules and pure paraffin (the measurement system is shown in Supplementary Fig. 6 and the test was carried out following the China National Standard GB/T 7314-2017). We first performed the tests at different temperatures, and the results are depicted in Fig. 4a. At room temperature, compared with pure paraffin, MTPCMs displayed significantly higher compressive yield strength, and this capacity was reinforced about 2.5 to 3.7 times with the increase of volume ratio in the sample. At the

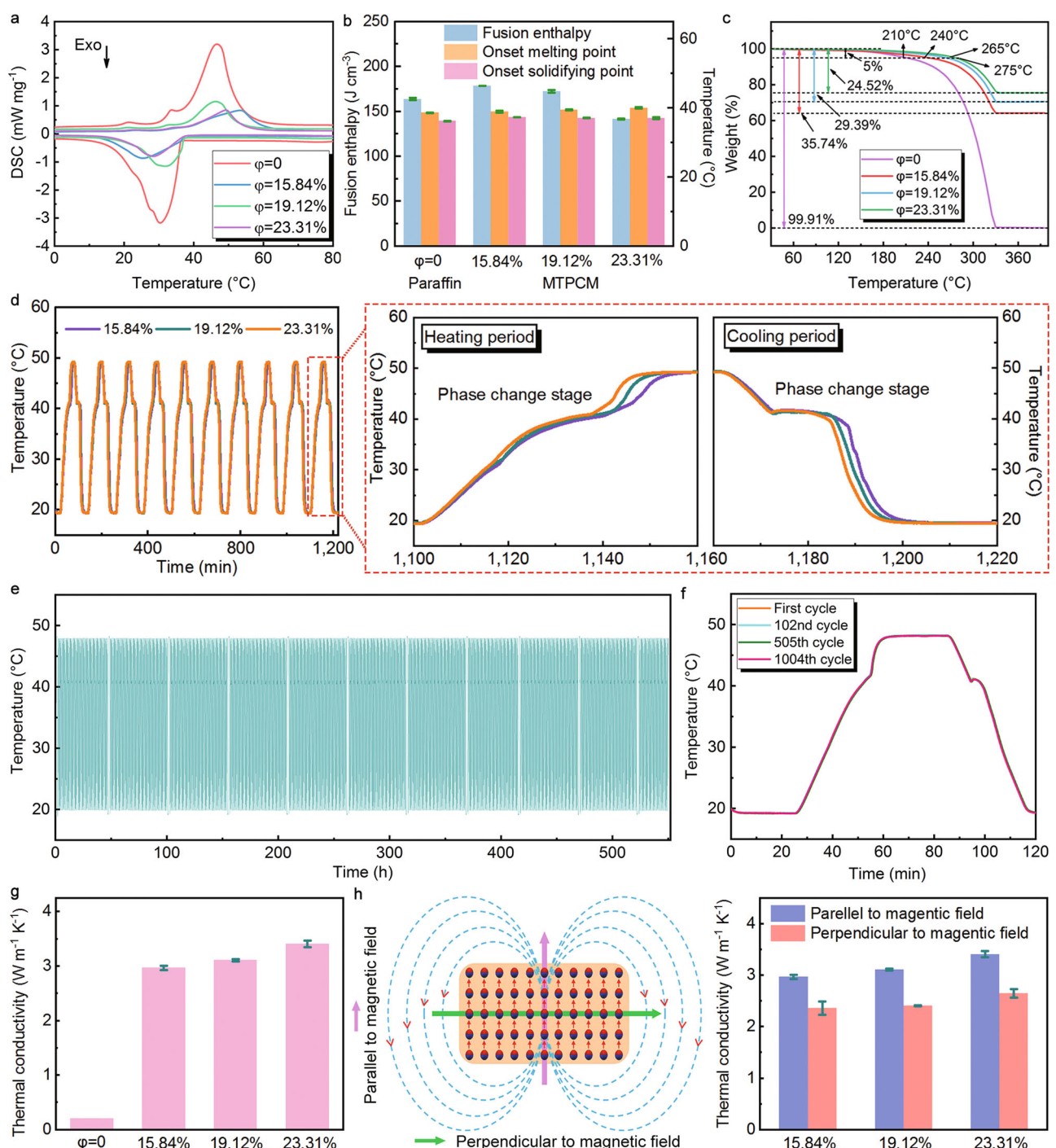

**Fig. 3 Thermal performance of magnetically tightened form-stable phase change materials (MTPCMs). a** DSC curves of paraffin and different volume ratio MTPCMs. **b** Comparison of fusion enthalpy, onset melting, and solidifying point of pristine paraffin and MTPCMs with different volume ratios. **c** TGA curves of paraffin and different volume ratio MTPCMs. **d** Temperature variations of MTPCM during the heat charging/discharging cycles and the magnified heating-freezing process in one period. **e** Over 1000 thermal cycles and **f** four selected typical heat charging/discharging cycles of MTPCM. **g** Thermal conductivities of pristine paraffin and MTPCMs with different volume ratios. **h** Schematic diagram of the directional alignment of NdFeB@Ag particles in paraffin matrix and the consequently anisotropic thermal conductivities of MTPCMs.

temperature above the melting point of paraffin, pure paraffin melted completely into a puddle of liquid, making it impossible to test at all. While to MTPCMs, they could hold their shape until being compressed by an external normal force. At this moment, MTPCMs continuously deformed and were ultimately compressed into a flattened cuboid. No leakage occurred in the whole process, suggesting even though being compressed at a high temperature, MTPCMs can still keep their leakage-proof ability

intact (Supplementary Movie 4). These can be observed from the state of typical compressed samples in the insets as well.

It seems to take the result for granted that the compressive strength of MTPCMs is better than pure paraffin. But the mechanism behind this phenomenon is worth discussing. By comparing the two materials, it can easily conclude that the magnetic NdFeB@Ag chains throughout the MTPCMs support the materials giving them a strong structure. They are the basis on

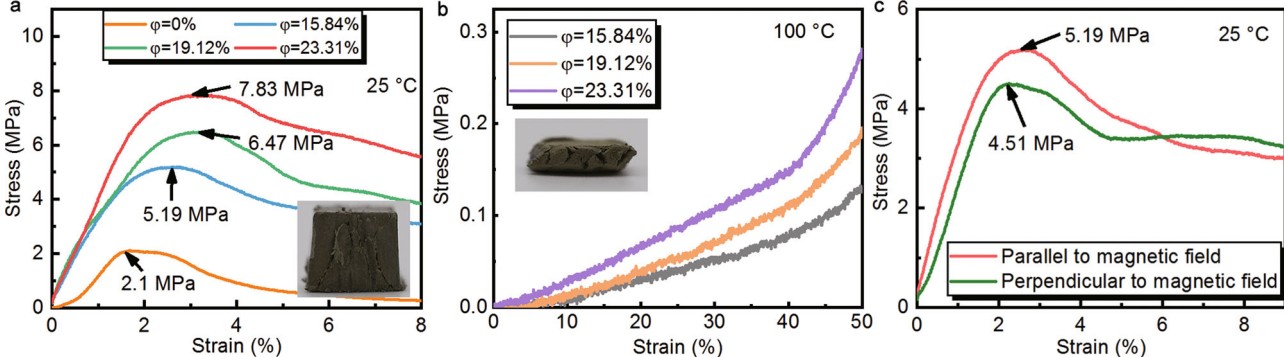

**Fig. 4 Mechanical performance of magnetically tightened form-stable phase change materials (MTPCMs). a** Compression stress-strain curves of pure paraffin and MTPCMs at the temperature below the melting point. **b** Compression stress-strain curves of MTPCMs at the temperature above the melting point. The insets are the typical compressed samples. Scale bars: 2 mm. **c** Anisotropic compression stress-strain curves of MTPCM at different directions.

which MTPCMs could own excellent compressive yield strength. However, in the high-temperature test, once the paraffin was melted, the magnetic chains yielded rapidly and the compressive yield strength of MTPCMs became almost zero. This indicates that paraffin also plays an important role in the compressive strength of MTPCMs. They tightly surround the NdFeB@Ag chains forming a protective layer that offers a lateral force when the sample is compressed. This helps the magnetic chains keep their original direction and continue to work as supports in the normal direction. Both the paraffin and magnetic chains are indispensable factors and under their combined actions, MTPCMs finally harvest a strong compressive strength. Based on the foregoing discussion, it should be also realized that MPTCMs have mechanical anisotropy. Thereby we measured the compressive strength of MTPCMs in different directions. As predicted, the sample has better compressive strength in the direction parallel to the magnetic field, originating from the anisotropic structure of MTPCMs. To sum up, compared to conventional PCMs, MTPCMs possess good mechanical performance at room temperature which could broaden their application to the field of green building.

**Energy conversion and storage of MTPCMs.** Employing PCMs in the areas of solar energy utilization, heat recovery, and power supply and demand regulation is all due to their large latent heat endowing them with big energy conversion and storage potentials[38–40]. Limited by the severe fluctuation of sunlight, the solar-thermal conversion system cannot support a persistent output. As a result, electric-thermal and thermo-electric conversion and storage systems have become promising alternatives[1,5,41]. On this occasion, we investigated the electricity-to-heat and heat-to-electricity conversion and storage performance of MTPCMs. As is known to all, organic paraffin is an insulator that is impossible to conduct electricity. However, as shown in Fig. 5a, the electrical conductivity of three MTPCM samples was $1.69 \times 10^4$, $2.25 \times 10^4$ and $2.83 \times 10^4$ S m$^{-1}$, respectively. The values are among the conductivity of metal materials ($\sigma > 10^3$ S m$^{-1}$), certifying MTPCMs are good conductors of electricity. To figure out the underlying mechanism of this change (from insulator to a good conductor), we adopted the control variable mode, measuring the electrical conductivity of unmagnetized composites and the post-magnetized mixture of NdFeB and paraffin, respectively. It was found that the latter material was insulated and the unmagnetized composites had lower electrical conductivities than MTPCMs (Fig. 5b). This result elucidates that the modified NdFeB@Ag particles is a dispensable term to enable MPTCMs to become conductive, and on this basis, the interconnected 3D structure strengthens this effect further

improving the electrical conductivity of MTPCMs. The electrical properties of MTPCMs are far beyond a high electrical conductivity. As discussed above, after being heated, NdFeB@Ag particles can reassemble building a new structure when the previous one is destroyed. With this feature, MTPCMs kept conductive regardless of encountering any deformations during the phase change process (Supplementary Movie 5). Owning such high electrical performance is profitable to the electric-thermal conversion and storage of MTPCMs, and this is confirmed by the following experiments. A voltage of 3 V was applied on the MTPCM to trigger its electro-heat conversion, and the current stayed at 0.27 A in the whole process. Based on the temperature evolution of the sample, it can be calculated that the phase change time lasted for about 61 s and the electricity-to-heat harvest efficiency of MTPCM was 78.45% (see Supplementary Fig. 7). Generally, the phase transition time is relatively long under low voltage and the heat dissipation from the sample to surroundings increases accordingly, leading to reduced efficiency[41]. Therefore, such high efficiency at a low voltage demonstrates the prominent electricity-to-heat conversion and storage capacity of MTPCMs. Besides, a thermal infrared test was also implemented to express the electro-thermal conversion performance of MTPCMs qualitatively (Fig. 5c). A power of 8.88 W was input to the heart-shaped sample, and it turned to about 65 °C in a short time of 30 s, suggesting the excellent performance of MTPCMs in the electricity-to-heat conversion and storage aspect again.

As another energy harvesting system, the heat-to-electricity conversion and storage is executed via two MTPCM samples and an in-between thermoelectric generator. Herein, the paraffin in MTPCMs was replaced by EBiInSn (eutectic alloy of Bi, In, and Sn) whose melting point is 60 °C to amplify the temperature gradient between the hot and cold sides. The commercial thermoelectric device was a semiconductor that can convert the heat in the MPTCM into electrical energy according to the principle of the Seeback effect[20,38]. As the heat source, the MTPCM was heated until the EBiInSn therein was melted. On account of the magnetic attraction feature of MTPCMs, the thermoelectric generator was tightly clamped having close contact with both two ends. We connected the system as a power supply to a self-programmed Bluetooth chip creating a hygrothermograph that can sensitively monitor the temperature and humidity of the local ambient in real-time (see Fig. 5d and Supplementary Movie 6). The evolution of temperature about the hot and cold side in the system and the corresponding voltage and current versus time are recorded in Fig. 5e. It can be seen that the MTPCMs realized a high and long-lasting temperature difference, and accordingly, the power supply duration was over 4 min, and the maximum voltage and current reached 0.39 V and 0.13 A. To

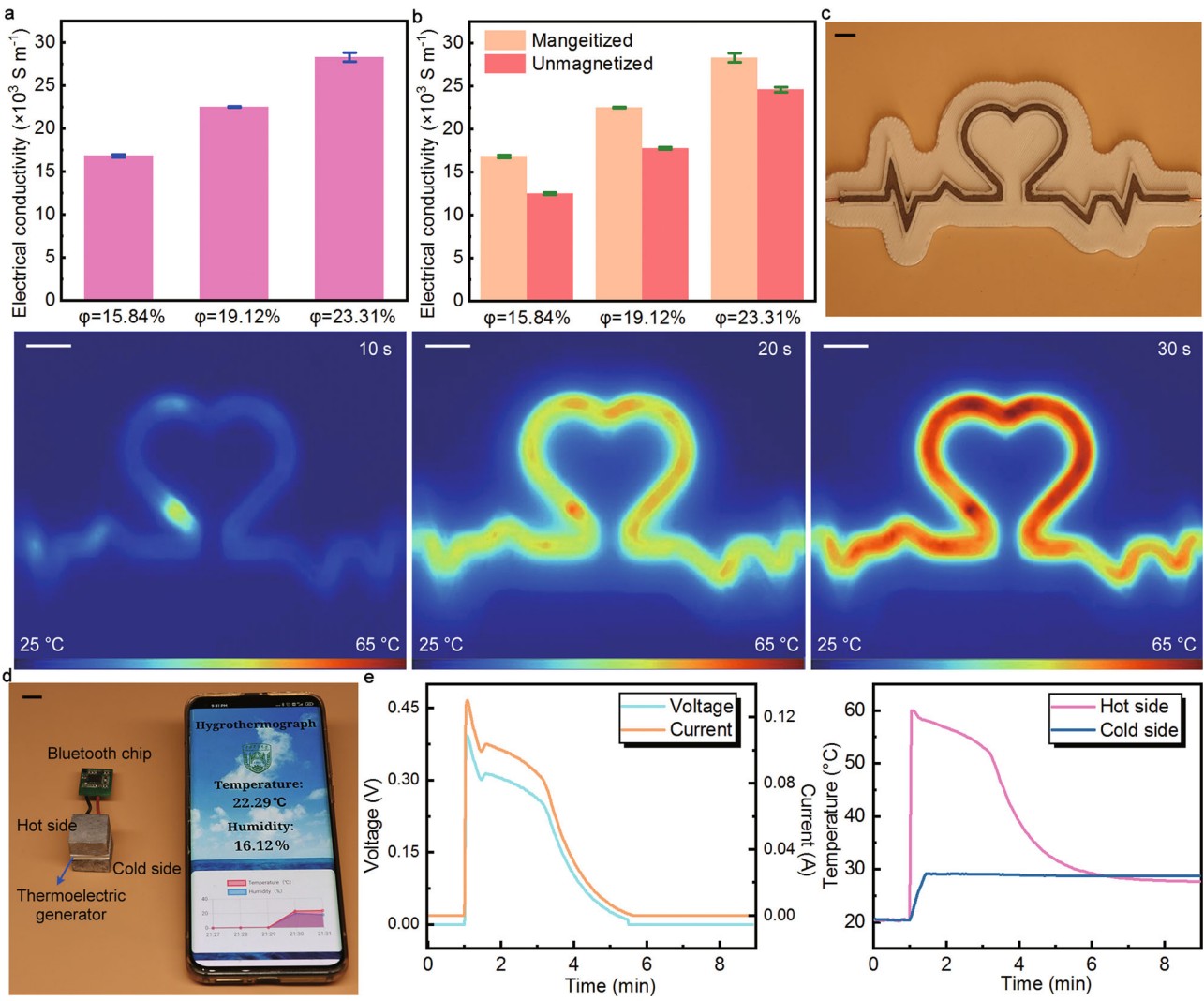

**Fig. 5 Energy conversion and storage applications of magnetically tightened form-stable phase change materials (MTPCMs). a** Electrical conductivities of MTPCMs with different volume ratios. **b** Electrical conductivities of MTPCM and unmagnetized composites. **c** Photo and thermal infrared images of MTPCMs during the electricity-to-heat conversion process. Scale bars: 5 mm. **d** Photo of the heat-to-electricity conversion and storage system. Scale bar: 10 mm. **e** Output voltage and current, and temperature evolution curves of the heat-to-electricity conversion and storage system.

sum up, a 4.47 J net electrical energy was harvested from a small MTPCM block (size: 20 × 20 × 10 mm). These data suggest the large latent heat and superior heat-to-electricity conversion and storage performance of MTPCMs.

**Thermal management of MTPCMs.** Being identified as an important application branch, thermal management demands PCMs master high thermal conductivity, large latent heat, and appropriate phase transition temperature at the same time, which are what exactly our MTPCMs highlight[42,43]. On top of that, most PCMs need containers and binders to prevent leakage and help them stick to the heat source[44–46]. However, introducing extra several layers between the PCM and heat source increases the thermal resistance. Combing the binders that are usually organic materials with very low thermal conductivity, the temperature control effect of PCMs will be greatly weakened. But for MTPCMs, this problem can be easily solved via their leakage-proof capacity and magnetism. As shown in Fig. 6a, two MTPCM blocks can directly attract to each other clamping the heat source between them with compact contact, which not only excludes the containers or binders between objects but also minimizes the interface contact thermal resistance through close contact. To illustrate the thermal

management capacity of MTPCMs, polyimide electric heating films were used as the heat source and they were sandwiched between two MTPCM and two paraffin blocks, respectively (Fig. 6b). The two structures were hung in the air to eliminate the influence of heat transfer between PCMs and the table. Unlike MTPCM blocks, it needed to drop some molten paraffin in the gap between two paraffin blocks to glue the heating film. When the electric heating films began to work, their temperature kept rising until reached the melting point of paraffin. From now on, the temperature of these two films went in different tendencies. Pure paraffin absorbed heat and melted to liquid, eventually felling off the heating film. While during this process, the MTPCM blocks were always attached to the film to absorb heat and held their shape unchanged (see Supplementary Movie 7). It can be seen from the temperature variations versus time that the two films presented the same temperature in the initial stage. From the transition point, the temperature of one film rose sharply, while the other film kept at about 40 °C for 5.8 min, and afterward its temperature also rose slowly.

The above is the case that the heat source is not magnetic. If itself is a magnetic substance or plating a magnetic layer on the surface of the heat source, the MTPCM can directly attach to it to carry out thermal management. For instance, as represented in

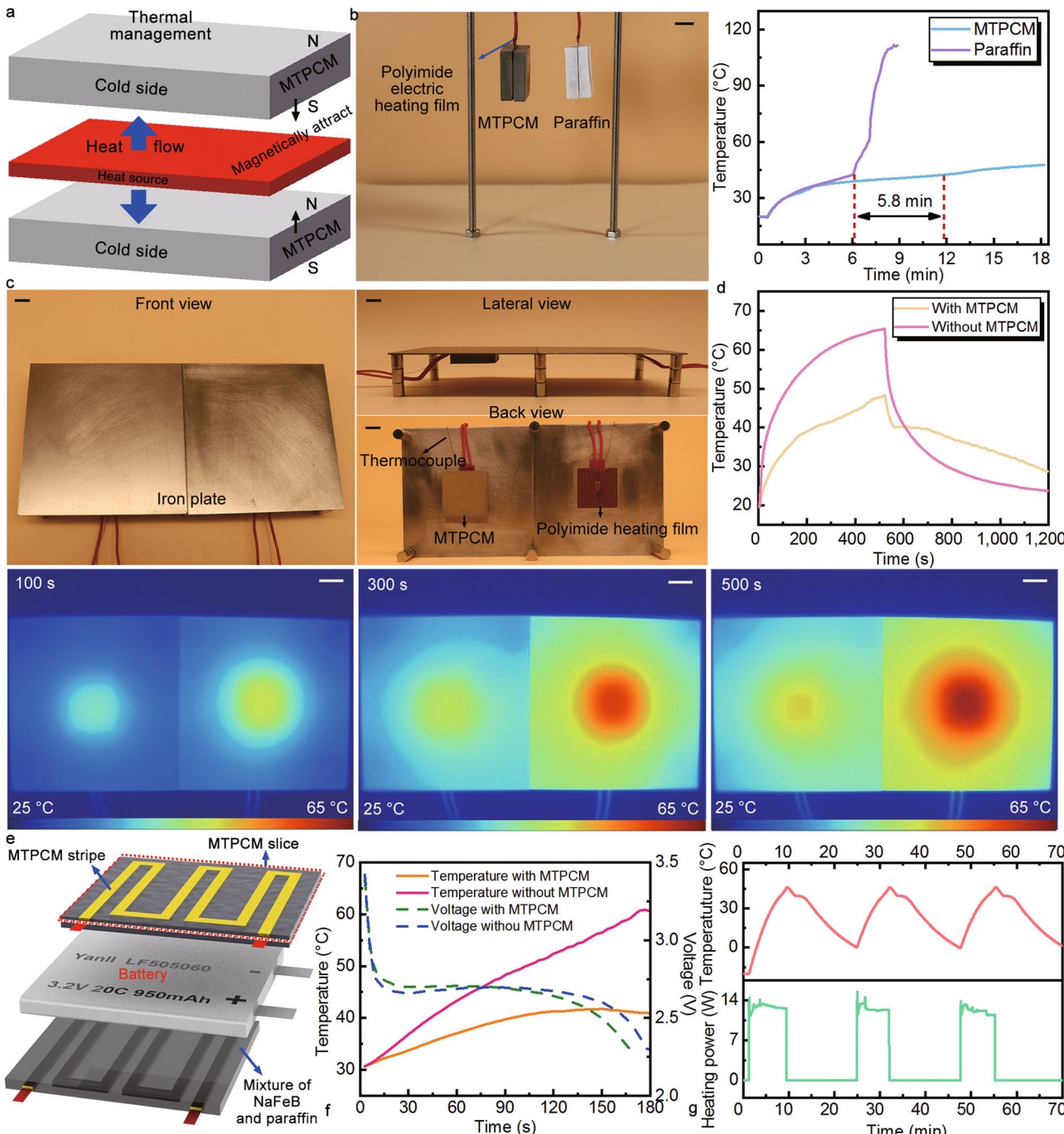

**Fig. 6 Thermal management applications of magnetically tightened form-stable phase change materials (MTPCMs).** **a** Schematic diagram of MTPCM blocks applied to the temperature control of a nonmagnetic heat source. **b** The corresponding photo and temperature evolution of MTPCM and paraffin blocks. Scale bar: 10 mm. **c** Photos about the temperature control of MTPCM on the magnetic heat source. Scale bars: 1 cm. **d** Temperature evolution and the corresponding thermal infrared images of the heat source with and without the MTPCM. Scale bars: 5 mm. **e** Schematic illustration of battery thermal management system. **f** Temperature and voltage evolution of lithium-ion battery during the discharging process. **g** Temperature variation of the battery and the heating power of one MTPCM slice under the low-temperature protection situation.

Fig. 6c, polyimide electric heating films stuck to the iron plates acting as the heat source. In virtue of the magnetism from the iron plates, the MTPCM block could directly contact the film for temperature control, and as a comparison, the other film did not take any active heat dissipation steps. When the heating started, the temperature of the film without MTPCM rose rapidly reaching 65.33 °C in 522 s under a power of 6 W, and once the power was cut off, its temperature dropped quickly to its initial value (Fig. 6d). Overall, the temperature of the heat source

fluctuated dramatically over a short period. But for the film using MTPCM block for thermal management, its temperature changed steadily and held at about 40 °C for 43 s and 73 s in the heating and cooling section, respectively. This difference in temperature change for the two heating films can also be discovered intuitively by the sequential thermal infrared pictures. The aforesaid two experiments prove that MTPCMs have superb thermal management capacity and this arises from the combination of their good thermal properties and magnetism.

Batteries face a severe problem in practical application, that is, their performance is heavily dependent on the temperature[47]. Too high and too low temperature or even non-uniform temperature distribution will depress their energy and power capacity making batteries suffer from capacity fade and short lifespan[48,49]. In this work, we engage MTPCMs for both heat dissipation at elevated temperature and low-temperature protection to regulate the working temperature of the flat-plate lithium-ion battery in a desirable range (Fig. 6e). The specific experimental details are recorded in the Methods section. When the battery discharged at a current of 17 A, its voltage dropped continuously until it reached the cut-off voltage at 168 s. While in this time the temperature increased slowly with the help of two MTPCM slices. It reached around 40 °C after 96 s and stayed at this point for 16 s, then moved to the highest temperature of 41.75 °C in the end. If the MTPCM slices were removed, for the same operation, the temperature in this process kept on rising and reached 60.85 °C (Fig. 6f). It can be calculated that MTPCMs reduce the battery temperature rise by 31.39% and this is attributed to the high thermal conductivity and large latent heat of MTPCMs. Towards the low-temperature situation, thanks to the excellent electrical properties, the MTPCM stripes can directly convert electricity to heat with high efficiency to warm up the battery from being frozen, forming an immediate protection scheme. In our experiment, the battery was cooled to −20 °C by the incubator. Once the outer MTPCM stripes were powered on and the electric power of about 12.82 W was supplied to each stripe, the battery was heated up and its temperature rose to 46.07 °C within 9.65 min. Even though the MTPCM stripes were powered off, the battery can still stay at the solidifying temperature of the MTPCM for 15 s and finally got a temperature of around 0 °C through one heating operation (Fig. 6g). After the battery cooled down, we powered on the MTPCM stripes again and it can be seen that the battery presented almost the same temperature variation, suggesting that the MTPCM slices own the cycling stability and can withstand the long and harsh working cycle conditions of the battery. The above heating and cooling experiments state that with prominent thermal performance and exclusive high electrical conductivity, MTPCMs have advantages in battery thermal management. They can control the battery temperature within normal values promptly, which is significant for the high performance and long working time of the battery.

## Discussion

Magnetism brings a strong binding force between magnetic particles. We utilize this effect and apply it to PCMs to solve their leakage issue. With this idea, the hard magnetic materials, NdFeB microparticles, were blended with paraffin. Given the low thermal conductivity of paraffin, NdFeB microparticles were firstly plated with a silver shell on their surfaces modifying them to NdFeB@Ag to enhance their thermal conduction. Through the magnetization operation, the NdFeB@Ag particles aggregated and aligned along the magnetic field forming an oriented 3D porous structure and the paraffin was carried in the micropores between particles. The magnetically-induced tightly connected particles lead to strong capillary force. Under this effect, the paraffin is stabilized in the structure synthesizing magnetically tightened form-stable phase change materials (MTPCMs). We studied and characterized this material thoroughly. It is found that the lowest volume ratio of NdFeB@Ag particles in MTPCMs is 15.84% and as long as the particles are more than this critical value, MTPCMs will own leakage-proof ability and shape stability. The directional alignment of NdFeB@Ag particles makes MTPCMs present strong magnetism. The surface magnetic flux density of the 15.84% MTPCM reaches over 30 mT, and the magnetism of MTPCMs

can maintain under a high temperature or over long periods. Hence, standard MTPCM modules can assemble and reconfigure into diverse robust shapes via magnetic attraction. Besides, through a simple heating operation process MTPCMs, it can also form into any shape while keeping the above abilities intact, and this is due to the interaction between magnetic particles as well. Hence, magnetism endows MTPCMs with magnetic assembly and conformal geometry features.

The 3D structure built by NdFeB@Ag particles is the base of MTPCMs. It not only allows MPTCMs to attain leakage-proof and magnetism capacities but also improves the performance of these materials. In the thermal aspect, the NdFeB@Ag particles just function as the supporting material and have no impact on paraffin. The values of onset melting and solidifying point for pure paraffin (38.65 and 36.23 °C) and the three different ratios MTPCMs (38.95 and 37.33 °C for 15.84%, 39.5 and 37.13 °C for 19.12% and 40.05 and 37.07 °C for 23.31% sample) are nearly the same. As for the fusion enthalpy, owing to the synergistic effect of the mixing ratio and density increase in MTPCMs, this parameter turns from 163.75 J cm$^{-3}$ for paraffin to 178.70, 172.59, and 141.63 J cm$^{-3}$ for the three MTPCMs. Thereby, it could be concluded that MTPCMs are not affected by the magnetic particles and still retain the proper melting/solidifying point and large latent heat. Except that, the TGA test presents that the addition of NdFeB@Ag particles delays the 5% weight loss temperature of MTPCMs by 30 to 65 °C, making MTPCMs possess excellent thermostability. As for the heat charging/discharging cycles, MTPCMs exhibit an efficient thermal storage/release rate. After over 1000 thermal cycles, the heat charging/discharging curve of MTPCM almost overlaps with the first cycle curve, demonstrating their superior thermal reliability and long service life in practical applications. Moreover, the interconnected structure provides heat transfer pathways, combined with the superior thermal conduction of NdFeB@Ag together enhancing the thermal conductivity of MTPCMs by 14–16 times than pristine paraffin, reaching 2.97, 3.11, and 3.41 W m$^{-1}$ K$^{-1}$ for the three different volume ratio MTPCMs. In the mechanical performance aspect, unlike conventional PCMs, the supporting structure reinforces the compressive strength of MTPCMs. Compared with the compressive yield strength of paraffin which is only 2.1 MPa, this value increases to at least 2.5 times reaching 5.19 MPa for MTPCMs at room temperature. Even at the temperature above the melting point of paraffin, MTPCMs can still maintain the leakage-proof ability under compression. Furthermore, magnetic particles' head-to-tail alignment triggers an anisotropic structure and consequently brings about a slight anisotropy in thermal and mechanical performance for MTPCMs.

With the above excellent properties, we apply MTPCMs in energy conversion and thermal management to evaluate their performance in practical applications. The 3D structure and silver shells of NdFeB@Ag also provide conduction paths for electricity, consequently endowing MTPCMs with the electrical conductivity of the order of 10$^4$ S m$^{-1}$. As a result, the electricity-to-heat harvest efficiency of MTPCM reaches as high as 78.45%, and a small 20 × 20 × 10 mm MTPCM block can harvest a 4.47 J net electrical energy from its latent heat. These illustrate the high energy conversion efficiency and energy harvesting ability of MTPCMs in electricity and heat conversion. While in the temperature control aspect, the MTPCMs have their unique advantages. The leakage-proof capacity and magnetism of MTPCMs allow them to directly attach to the hot object for temperature control without using containers and binders, which greatly reduces the thermal resistance between PCMs and heat sources. On top of that, MTPCMs perform well in battery thermal management. When the battery discharges at a current of 17A, the MTPCM function its heat dissipation role reducing the

temperature rise of the battery by 31.39%. While at low temperature, the MTPCM protects the battery by converting electricity to heat, regulating its temperature from −20 °C to keep it above 0 °C. In conclusion, introducing hard magnetic particles to PCMs proposes an efficient route to tackle the liquid leakage and low thermal conductivity issues of PCMs, and the accompanying-generated series of properties in PCMs make them preponderant in practical applications.

## Methods

**Materials**. NdFeB microparticles were purchased from Magnequench International. LLC. Silver nitrate (AgNO₃) was bought from Aladdin Industrial Corporation. Poly-vinylpyrrolidone (PVP, average molecular weight = 8000), D- Glucose, and ammonium hydroxide solution were provided by Macklin Biochemical Co., Ltd.

**Characterizations and property measurements**. SEM and EDX images were observed by QUANTA FEG 250, the image resolution of this microscope was 2.5 nm at 30 kV. XRD patterns were tested using the Bruker D8 Focus X-ray diffractometer, the angular deviation of all peaks in the full spectrum range is less than ±0.01°. Magnetic hysteresis loops were obtained by a cryogen-free cryocooler-based physical property measurement system (VersaLab) from Quantum Design Inc, the resolution of temperature measurement was 0.5 K and the root mean square sensitivity was less than $10^{-5}$ emu for this equipment. In our experiment, the magnetization and demagnetization rate was set as 100 Oe s⁻¹ with the magnetic field change of up to 30 kOe. The surface magnetic flux density was measured by Gaussmeter. DSC measurement was implemented on NETZSCH DSC 200F3 Maia equipment with a ramp rate of 10 K min⁻¹. The resolution of this instrument was 0.1 K when testing the melting and solidifying point and it became 1% when testing the fusion enthalpy. TGA was conducted by NETZSCH TG 209F3 instrument with a heating rate of 10 K min⁻¹ under the nitrogen atmosphere, and the resolution of this instrument was 0.1 μg. In a complete heating/freezing process, the sample stayed at 20 °C for 30 min. Then it was heated to 50 °C in 30 min and maintained at this temperature for 30 min. After this stage, the sample was cooled to 20 °C and kept at this temperature point for 30 min. Regarding the over 1,000 thermal cycles, we adjusted the heating/cooling rate and the constant temperature time to improve the cycle efficiency. In a complete heating/cooling process, the temperature program was set to stay at 20 °C for 5 min, then it began to heat up to 50 °C in 8 min and stayed at this point for 10 min. In the cooling phase, the temperature cooled to 20 °C in 8 min and stayed at this point for 5 min. Besides, the heating/cooling process was performed once followed the aforementioned charging/discharging program after about every 100 cycles (i.e. stayed at 20 °C for 30 min, heated to 50 °C in 30 min and stayed at this point for 30 min, cooled down to 20 °C in 30 min and stayed at this point for 30 min). Finally, we cycled 1054 times. Thermal conductivity was calculated by the equation $\lambda = a \times C_p \times \rho$, where $a$ represents the thermal diffusivity, $C_p$ is specific heat capacity, and $\rho$ is density. The thermal diffusivity was measured by the NETZSCH LFA 467 instrument and density was acquired using the METTLER TOLEDO XSE105 analytical balance with XS/XP-Ana Density Kit. The error of the LFA 467 instrument was 3%. For specific heat capacity, it was tested by the abovementioned DSC equipment, and its error was 3–5% when measuring the specific heat capacity. The mechanical property of MTPCMs was measured using a universal testing machine (MTS-E45.105) equipped with an incubator, and the compression rate was set as 2 mm min⁻¹ during the test and the error of the machine was 0.5%. Electrical conductivity was measured by a standard four-point method. The sample was filled in a tailored groove, a voltage was applied and the resistance was measured by a micro ohmmeter (Agilent 34420 A). The error of the Agilent 34420 A was 0.003%. The electrical conductivity was calculated using $\sigma = s/R \times l$, where $s$ is the cross-sectional area, $R$ is the resistance of the sample, and $l$ is the length of the groove.

**Surface modification of NdFeB**. The surface modification of NdFeB to NdFeB@Ag was implemented by in situ chemical plating processing. Typically, weight ratios of several reagents were selected as NdFeB:AgNO₃ = 2:1, PVP: AgNO₃ = 1:1 and glucose: AgNO₃ = 2:1. Firstly, PVP powder (40 g) together with NdFeB particles (20 g) was added to adequate deionized water (DI). After PVP was thoroughly dissolved in DI, mechanical agitation was employed to the suspension for 20 min to make sure every NdFeB particle was covered by a PVP film. Then, the suspension was kept still for a few minutes and the supernatant was poured off to obtain the precipitated NdFeB particles. Next, the silver ammonia solution was prepared by dropping ammonia hydroxide in the AgNO₃ solution (which was prepared by dissolving 40 g AgNO₃ in about 60 mL DI) until it turned transparent, and NdFeB particles were resuspended in silver ammonia solution for the following plating process. The suspension was first mechanically agitated for 10 min and afterward, it was transferred to the heating platform. As the reductant of the reaction, glucose solution (which was prepared by dissolving 80 g glucose powder in about 60 mL DI) was produced by dissolving glucose in DI. The optimum temperature of the silver plating reaction was above 60 °C. Until the suspension was heated to a relatively high temperature, glucose solution was added and the mechanical agitation was turned on. During the reaction, it can be observed that the suspension became yellow slurry in a short time indicating the successful

reduction of silver ions. The silver plating reaction was done in about 15 min and the synthesized NdFeB@Ag particles were extracted, rinsed, and dried in a vacuum oven. The weight of each reagent involved in the reaction can be scaled up in proportion. Consequently, the production of NdFeB@Ag can be increased.

**Preparation of MTPCMs**. NdFeB@Ag particles can be easily dispersed in melted paraffin by manually stirring. To fabricate MTPCMs, a tailored rectangle mold with an electrical heating function was used to keep the composite liquid during the magnetization process, which enables NdFeB@Ag particles to rotate and align along the magnetic field. The mold filled with the composite was placed on the sample stage and a magnetic field with 2 T magnetic flux density was exerted to the composite for 30–60 s. After magnetization, the heating function was turned off and a moderate pressure was applied to the MTPCM until it solidifies. The pressure could eliminate the porosity in MTPCMs, which is a common issue in composite materials. For as-prepared MTPCMs, they can be processed to standard modules by a wire cutting machine. It could be concluded that the magnetization process cost a very short time, and the time was mainly spent on heating and cooling the materials. Therefore, the efficiency of fabricating MTPCMs could be improved by customizing a larger mold and employing more efficient heating and cooling equipment.

**Thermal management of battery**. The commercial flat-plate lithium-ion battery had a nominal capacity of 950 mAh, a nominal voltage of 3.2 V, and a maximum continuous discharge rate of 20 C. To obtain a better electro-heat conversion effect, the MTPCM was fabricated to a stripe with 5 mm width × 1.5 mm thickness × 20 mm length. Then the stripe was designed to be embedded in a 60 mm × 50 mm × 3 mm matrix with a serpentine shape to acquire uniform temperature distribution in the heating/cooling process. The matrix was a mixture consisting of NdFeB particles and paraffin. Since the phase change material in matrix and stripe was both paraffin and paraffin was stabilized by magnetism in the two materials, the as-prepared slices were called MTPCM slices. The MTPCM slices can attract each other clamping the battery between them to implement the heating and cooling experiment. As shown in Supplementary Fig. 8, the battery was discharged by a battery testing system in the cooling process, and the MTPCM stripes were heated by electricity in the heating process.

## Data availability

All relevant data generated or analyzed during this study are available in this published article and its supplementary information files. Source data are provided with this paper.

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

## Acknowledgements

The authors would like to acknowledge the National Natural Science Foundation of China (NSFC) (Grant No.52076213), the 2115 Talent Development Program of China Agricultural University, and the NSFC Key Project (Grant No.91748206) for the financial coverage of this work.

## Author contributions

Y.L. and D.Y. designed and conducted the experiments. J.L. and Z.H. supervised the investigations. H.D. did lots of work in the preliminary experiment. J.L. programmed the hygrothermograph system. H.Z. performed the magnetic characterization. L.C. helped with the sample fabrication. Y.L. drafted the manuscript with inputs from all other authors. All the authors agreed on the final manuscript.

## Competing interests

The authors declare no competing interests.
