## [Peer Review File · Nature Communications]

REVIEWER COMMENTS

Reviewer #1 (Remarks to the Author):

In this manuscript, the authors proposed a novel form-stable method and prepared magnetically tightened form-stable PCMs (MTPCMs). The MTPCMs achieved multifunctions of leakage-proof, dynamic assembly and morphological reconfiguration. This is a very creative work, and I have a few questions.

1. In Figure 1a and its description, you mentioned the orientation of NdFeB@Ag particles. Did axial or radial magnetization be used in magnetization? If magnetized in all directions, is it better to form a 3D magnetic structure?
2. You mentioned “magnetically tightened” in the title, while in experimental section, you mentioned a moderate pressure was applied to the MTPCMs samples. Is the tightening caused by magnetic force or by the pressure of the preparing process? Whether the same pressure was applied to the paraffin sample as a control?
3. Existing literatures on form-stable PCMs and introducing magnetic materials into PCMs in recent years should be cited in Introduction.

Reviewer #2 (Remarks to the Author):

This paper is highly interesting and contains good technical information related to the form-stable/shape stabilization of the PCM under the influence of magnetic field which has achieved multifunctional abilities including leakage-proof, dynamic assembly and morphological reconfiguration, superior high thermal and electrical conductivities, and prominent compressive strength.

I appreciate the authors for having presented the information about the developed FSPCM in terms of the innovation, materials used, methodology, scientific reasoning and justification, discussion of the results, graphical presentation of the results and application perspectives. These aspects are considered to be most essential parts of the study.

The paper can be accepted for publication, provided the following points are to be considered for revision of the paper:

1. There was no mention about which paraffin PCM being used for experiments and why it was selected when other organic PCMs are available? There are a number of paraffin wax PCMs are available based on the number of carbon atoms. So, please elaborate on this.
2. Characterization methods should be elaborated by including the specifications of each characterization equipment along with their accuracy.

3. Uncertainty/errors associated with the measurements must also be included in this study as this is purely an experimental work.
4. The heating/cooling rate of DSC being used for measuring the phase change characteristics of the MTPCMs was too high in fact. This might have not resulted in proper phase transition characteristics of the developed PCM. Please justify.
5. The effects due to supercooling and hysteresis during DSC measurements may also be the factors affecting the latent heat enthalpy and phase change temperature values being obtained for the PCM samples. Authors can include discussions on these phenomena too.
6. Page 12, lines 219-221: MTPCMs were only tested for 10 cycles. With this, claiming of thermal-cycle reliability and durability of MTPCMs to be outstanding seems to be logically incorrect. That too from the application point of view. In order to appreciate the feasibility of the MTPCMs, at least 1000 thermal cycling (charging and discharging cycle) tests have to be performed. This test is mandatory for proving the thermal stability, shape stability and thermal storage performance of the prepared MTPCMs. Authors are strongly suggested to carry out this test and include the results in the updated version of this paper.
7. Compressive strength test was not performed properly. What was the targeted compressive strength for the prepared MTPCMs? Also, there was no mention about which International Standards/Code of Practices were followed while conducting this test. What was the compressive strength for the pristine Specimen and how much the deviation that was obtained in the value when compared to the MTPCMs specimens? Please address this carefully and include more discussion of the results, in place of presenting only the results.
8. There was no discussion related to the specific heat capacity measurements for MTPCMs, which is highly essential as they play a vital role in phase change kinetics. The thermal properties related findings made on MTPCMs has to gel well with the thermal conductivity and specific heat measurements. Else, it becomes a less convincing study. Please address this carefully.
9. For the developed form-stable MTPCMs, it would be wise to report their encapsulation ratio, encapsulation efficiency and thermal storage capability. Without these parameters, it becomes very difficult to substantiate the usefulness of the prepared PCMs.
10. Conclusion must be well supported with the data.

Reviewer #3 (Remarks to the Author):

This paper presents "Magnetically tightened form-stable phase change materials with modular assembly and geometric conformality features". I am convinced that such preparing magnetically tightened form-stable phase change materials in this article have a guiding significance for the preparation of PCM composites. The authors did a solid job. While the article has significant shortfalls as presented below. In my opinion, the following comments must be addressed before considering the publication.

The summary of the article does not adequately reflect the work done.

10 thermal cycles are not enough for new materials. Authors are required to include their thermal and chemical analysis after at least 200 thermal cycles.

Leakage tests are one of the most important analyses tests for the PCM composites. Could you please give information about the leakage test time applied in this study? Even if the PCM composite leakage, it is difficult to distinguish with the naked eye in short times. Specifically, the prepared samples need to be heated to 50°C on the filter paper for approximately 2 hours to observe whether there are traces of liquid leakage on the filter paper during the phase transition.

Also, not only a digital image but also DSC, SEM, FTIR, and TEM image is necessary to demonstrate if the PCM leaked during the leakage test.

Only EDS spectra cannot prove “the interactions of magnetic material and PCM during the preparation of form stable composites are physical and no chemical interaction occurs.” More measurements such as Raman, FTIR are needed to certify this statement.

Reviewer #1:

In this manuscript, the authors proposed a novel form-stable method and prepared magnetically tightened

form-stable PCMs (MTPCMs). The MTPCMs achieved multifunctions of leakage-proof, dynamic assembly and morphological reconfiguration. This is a very creative work, and I have a few questions.

Reply: We appreciate your great encouraging comments on our work. We have modified our manuscript according to your very helpful suggestions. Our responses are listed below and the revised portion is marked in red in the manuscript. Again, thank you very much for helping us improve our manuscript.

1. In Figure 1a and its description, you mentioned the orientation of NdFeB@Ag particles. Did axial or radial magnetization be used in magnetization? If magnetized in all directions, is it better to form a 3D magnetic structure?

Reply: Thank you for your question. In our experiment, the MTPCM samples are magnetized under axial or radial magnetization during the magnetization process, which is easily controlled just by adjusting the magnetization direction. For magnetic particles, their intrinsic magnetic moments compel the particles to rotate and align along the magnetization direction. Due to the strong interaction between the magnetized particles, the alignment of microparticles could induce MTPCMs' mechanical property change, resulting in a remarkable stiffening effect even for the liquid PCM matrix without triggering by external pressure. The magnetization could enable MTPCM to behave like a permanent magnet (such as a high surface magnetic flux density of 31 mT for a thickness of 10 mm, see Fig.2e), which is essential to prepare a self-assembly MTPCM block with a complex shape (see Fig.1e). It is exciting to find that MTPCM under liquid state could be easily reshaped to disorder alignment of magnetic particles and disable its macroscopical magnetic performance. In this case, the MTPCM could remain the stable 3D magnetic structure due to the strong interaction between the magnetized particles. The MTPCM could be recovered its macroscopical magnetic performance through remagnetization.

2. You mentioned “magnetically tightened” in the title, while in experimental section, you mentioned a moderate pressure was applied to the MTPCMs samples. Is the tightening caused by magnetic force or by the pressure of the preparing process? Whether the same pressure was applied to the paraffin sample as a control?

Reply: We thank the reviewer for the kind question. It needs to elucidate that the tightening effect in MTPCMs is only caused by the magnetic force. We use the expression “magnetically tightened” to describe the underlying mechanism of shape-stable and leakage-proof features in MTPCMs. The magnetic attraction between NdFeB@Ag particles in MTPCMs is the main reason why our materials own these two features. The magnetism provides a force to strain the particles together, subsequently produce the capillary force between magnetic particles and finally immobilize the paraffin in MTPCMs. To describe this effect of magnetism more vividly, we use the word “tightened”. As for the moderate pressure in the sample preparation process, it is just a conventional way to better shape the material for performance testing after being magnetized. And to avoid the paraffin from being forced out under excessive pressure, we emphasize the “moderate pressure” in the article. At last, we have to point out that applying pressure is just a conventional but not a mandatory requirement, it can help make the fabricated material more condensed and easy to shape the material.

3. Existing literatures on form-stable PCMs and introducing magnetic materials into PCMs in recent years

should be cited in Introduction.

Reply: Thank you very much for the reminding. Based on your advice, we have carefully searched more papers about form-stable PCMs and cited them in the Introduction section to further elucidate the advantages and existing problems about form-stable PCMs. Besides, we also cited two papers that encapsulate PCMs in nanocapsules. One paper is a review which elaborates the encapsulation of PCMs and the other paper brings PCMs to biomedical applications which is very rare in PCM research. As for introducing magnetic materials into PCMs, we cited a paper in our article. Meanwhile, we have to say that the related research is relatively less in this field and current papers are usually adding magnetic particles to make use of their special effects (such as magnetothermal effect, vibration heat production effect, etc.) to carry out thermal energy conversion and management. Up to now, there is no paper using magnetic particles as the supporting material to prepare form-stable PCMs.

The cited papers have been highlighted in the manuscript:

7. Tang Z, Gao H, Chen X, Zhang Y, Li A, Wang G. Advanced multifunctional composite phase change materials based on photo-responsive materials. *Nano Energy* 80, 105454 (2021).
8. Chen X, Cheng P, Tang Z, Xu X, Gao H, Wang G. Carbon-Based Composite Phase Change Materials for Thermal Energy Storage, Transfer, and Conversion. *Adv Sci (Weinh)* 8, 2001274 (2021).
10. Aftab W, et al. Highly efficient solar-thermal storage coating based on phosphorene encapsulated phase change materials. *Energy Storage Materials* 32, 199-207 (2020).
11. Shchukina EM, Graham M, Zheng Z, Shchukin DG. Nanoencapsulation of phase change materials for advanced thermal energy storage systems. *Chem Soc Rev* 47, 4156-4175 (2018).
12. Aftab W, et al. Polyurethane-based flexible and conductive phase change composites for energy conversion and storage. *Energy Storage Materials* 20, 401-409 (2019).
14. Aftab W, Usman A, Shi J, Yuan K, Qin M, Zou R. Phase change material-integrated latent heat storage systems for sustainable energy solutions. *Energy & Environmental Science* 14, 4268-4291 (2021).
15. Cui J, et al. Ultra-Stable Phase Change Coatings by Self-Cross-Linkable Reactive Poly(ethylene glycol) and MWCNTs. *Advanced Functional Materials* n/a, 2108000 (2021).
17. Wei P, Cipriani CE, Pentzer EB. Thermal energy regulation with 3D printed polymer-phase change material composites. *Matter* 4, 1975-1989 (2021).
18. Chen X, Gao H, Tang Z, Dong W, Li A, Wang G. Optimization strategies of composite phase change materials for thermal energy storage, transfer, conversion and utilization. *Energy & Environmental Science* 13, 4498-4535 (2020).
25. Qiu J, Huo D, Xue J, Zhu G, Liu H, Xia Y. Encapsulation of a Phase-Change Material in Nanocapsules with a Well-Defined Hole in the Wall for the Controlled Release of Drugs. *Angew Chem Int Ed Engl* 58,

10606-10611 (2019).

27. Wu S, Li T, Zhang Z-Y, Li T, Wang R. Photoswitchable phase change materials for unconventional thermal energy storage and upgrade. *Matter* 4, 3385-3399 (2021).

29. Chen X, et al. Carbon nanotube bundles assembled flexible hierarchical framework based phase change material composites for thermal energy harvesting and thermotherapy. *Energy Storage Materials* 26, 129-137 (2020).

33. Liu L, Hu J, Fan X, Zhang Y, Zhang S, Tang B. Phase change materials with Fe₃O₄/GO three-dimensional network structure for acoustic-thermal energy conversion and management. *Chemical Engineering Journal* 426, 130789 (2021).

Reviewer #2:

This paper is highly interesting and contains good technical information related to the form-stable/shape stabilization of the PCM under the influence of magnetic field which has achieved multifunctional abilities including leakage-proof, dynamic assembly and morphological reconfiguration, superior high thermal and electrical conductivities, and prominent compressive strength. I appreciate the authors for having presented the information about the developed FSPCM in terms of the innovation, materials used, methodology, scientific reasoning and justification, discussion of the results, graphical presentation of the results and application perspectives. These aspects are considered to be most essential parts of the study. The paper can be accepted for publication, provided the following points are to be considered for revision of the paper:

Reply: We appreciate the reviewer for kindly helping review our work and offering us very constructive and critical thoughts. We have carefully revised the manuscript according to your comments, which will help improve its clarity and quality. Below is a list of the point-by-point responses to address each of the comments and the corresponding changes are highlighted in the manuscript text, as well.

1. There was no mention about which paraffin PCM being used for experiments and why it was selected when other organic PCMs are available? There are a number of paraffin wax PCMs are available based on the number of carbon atoms. So, please elaborate on this.

Reply: Thank you for the question. The method we developed has a generalized purpose. As the first-ever trial in the field, the paraffin we used in the present experiment is not specially selected. The paraffin was bought from Shang Hai Joule Wax. Ltd and we often bought paraffin from this company. As selected paraffin as the phase change material, it is just because paraffin is a widely used phase change material. Except for paraffin, other PCMs not limited to organic PCMs can also be used to fabricate magnetically tightened form-stable phase change materials. For instance, stearic acid, polyethylene glycol, octadecanol, and inorganic low-melting-point alloy can all be chose as phase change materials. To prove this, we take the above materials as PCM to fabricate MTPCMs, and their melting point is 69, 65.9, 55.7, and 60 °C, so we heat these MTPCMs at 80 °C for 30 min. The experimental result is shown below, and it can be seen that these MTPCMs also own leakage-proof ability.

2. Characterization methods should be elaborated by including the specifications of each characterization equipment along with their accuracy.

Reply: Thank you for your valuable advice. We completely agree with your suggestion and add the information of every piece of equipment's accuracy in the characterizations and property measurements section. The revised text is shown as follows:

“SEM and EDX images were observed by QUANTA FEG 250, the image resolution of this microscope was 2.5 nm at 30 kV. XRD patterns were tested using the Bruker D8 Focus X-ray diffractometer, the angular deviation of all peaks in the full spectrum range is less than $\pm 0.01^\circ$. Magnetic hysteresis loops were obtained by a cryogen-free cryocooler-based physical property measurement system (VersaLab) from Quantum Design Inc, the resolution of temperature measurement was 0.5 K and the root mean square sensitivity was less than 10^{-5} emu for this equipment. In our experiment, the magnetization and demagnetization rate was set as 100 Oe s^{-1} with the magnetic field change of up to 30 kOe. The surface magnetic flux density was measured by Gaussmeter. DSC measurement was implemented on NETZSCH DSC 200F3 Maia equipment with a ramp rate of 10 K s^{-1} . The resolution of this instrument was 0.1 K when testing the melting and solidifying point and it became 1 % when testing the fusion enthalpy. TGA was conducted by NETZSCH TG 209F3 instrument with a heating rate of 10 K min^{-1} under the nitrogen atmosphere, and the resolution of this instrument was 0.1 μg .

Thermal conductivity was calculated by the equation $\lambda = a \times C_p \times \rho$, where a represents the thermal diffusivity, C_p is specific heat capacity, and ρ is density. The thermal diffusivity was measured by the NETZSCH LFA 467 instrument and density was acquired using the METTLER TOLEDO XSE105 analytical balance with XS/XP-Ana Density Kit. The error of the LFA 467 instrument was 3 %. For specific heat capacity, it was tested by the abovementioned DSC equipment, and its error was 3 %~5 % when measuring the specific heat capacity. The mechanical property of MTPCMs was measured using a universal testing machine (MTS-E45.105) equipped with an incubator, and the compression rate was set as 2 mm min^{-1} during the test and the error of the machine was 0.5 %. Electrical conductivity was measured by a standard four-point method. The sample was filled in a tailored groove, a voltage was applied and the resistance was measured by a micro ohmmeter (Agilent 34420A). The error of the Agilent 34420A was 0.003 %. The electrical conductivity was calculated using $\sigma = s/R \times l$, where s is the cross-sectional area, R is the resistance of the sample, and l is the length of the groove.”

As for the test specifications, these instruments including SEM, XRD, VSM (VersaLab), Gaussmeter, DSC, TGA, LFA, universal testing machine, and the micro ohmmeter are commercial instruments, and their

operations are fixed and no special instructions are required in usual practice.

3. Uncertainty/errors associated with the measurements must also be included in this study as this is purely an experimental work.

Reply: Thank you for your valuable suggestion. We have considered the uncertainty of the data and tried our best to calculate the mean and standard deviation of the data. For example, in the DSC test, the heating/cooling process was cycled 3 times to calculate the mean and standard deviation of the onset melting/solidifying point and fusion enthalpy of samples. For the density, specific heat capacity, and thermal diffusivity, we also repeated 3 times to calculate the mean and standard deviation of the thermal conductivity of samples. The error bars of the above parameters are shown in Fig. 3b, 3c, 3e, and 3f, respectively. In addition, the electrical conductivity test was also repeated at least 5 times to acquire the mean and standard deviation of this parameter, and we displayed the error bar in Fig. 5a and 5b. As for the mechanical test, it is a destructive test that the sample will be damaged after being compressed. Hence, the test is only done once. For other experiments, such as the heat charging/discharging experiment, energy conversion, and storage experiments, and thermal management experiments, the parameters are all varied versus time and it needs to monitor and record their real-time values. Therefore, these data did not have the mean and standard deviation.

4. The heating/cooling rate of DSC being used for measuring the phase change characteristics of the MTPCMs was too high in fact. This might have not resulted in proper phase transition characteristics of the developed PCM. Please justify.

Reply: Thank you for the reminder question which did help us better think deeply. In our DSC test, the heating/cooling rate was set as 10 K/min which is a commonly used heating/cooling rate. Given your advice, we take the 15.84% MTPCM as a sample and test its DSC curves using three different heating/cooling rates to justify the impact of the scanning rate on the DSC results. It is necessary to point out that the melting/solidifying point of a material is the starting point of the melting/solidification peak on DSC curves, and the fusion enthalpy is the area of the melting peak (Please see detailed discussion in Ref.: Hohne GWH, et al. The Temperature Calibration of Scanning Calorimeters. *Thermochim Acta*. 1990(160):1-12.). In our paper, these three values are calculated by the software in the DSC instrument. The melting point is considered to be the intersection point of the tangent line and the horizontal line at the starting position of the melting peak on the melting curve. While for the freezing curve, observing the solidification peak from right to left, as the starting point of the peak is very sharp, it is generally considered that the starting point is the solidifying point.

We listed the DSC curves and corresponding parameters in the below table. It can be seen that although the scanning rate affects the DSC curve, our results indicate that the heating/cooling rate has a negligible impact on the onset melting/solidifying point and the fusion enthalpy of the MTPCM sample. Thus, the selected DSC scanning rate of 10 K/min could ensure the reliability and repeatability of the results.

15.84% MTPPCM	Onset melting point/ °C	Onset solidifying point/ °C	Fusion enthalpy/J/cm ³
5 K/min	39.6	39.6	183.62
10 K/min	39.6	39.1	183.99
15 K/min	40.3	38.4	184.62

5. The effects due to supercooling and hysteresis during DSC measurements may also be the factors affecting the latent heat enthalpy and phase change temperature values being obtained for the PCM samples. Authors can include discussions on these phenomena too.

Reply: Thank you for your question which is very careful with the DSC test. To this issue, we consulted the engineer of NETZSCH and read the specification to figure out the fundamental principle of the DSC instrument. The DSC test needs to set up the test program firstly. In this process, we must pick on the STC function which is the abbreviation of “sample temperature control”. This function can ensure there is no supercooling or hysteresis between the furnace of the DSC instrument and the sample. The DSC instrument is a mature commercial instrument to test the melting/solidifying and latent heat of the materials and it needs to be calibrated every year. After calibration, the test results are reliable and one can believe that there is no supercooling or hysteresis during measurements.

6. Page 12, lines 219-221: MTPCMs were only tested for 10 cycles. With this, claiming of thermal-cycle reliability and durability of MTPCMs to be outstanding seems to be logically incorrect. That too from the application point of view. In order to appreciate the feasibility of the MTPCMs, at least 1000 thermal cycling (charging and discharging cycle) tests have to be performed. This test is mandatory for proving the thermal stability, shape stability and thermal storage performance of the prepared MTPCMs. Authors are strongly suggested to carry out this test and include the results in the updated version of this paper.

Reply: Thank you so much for your advice. We completely agree with your kind suggestion. Over the past few weeks, we have been intensively working on this issue and tremendous additional experiments were performed. During the test, we took the 23.31% MTPPCM as the typical sample and cycle the heat charging/discharging process for 1054 times which lasted nearly one month. And to improve the efficiency, we increased the heating/cooling rate of cycles and after about every 100 cycles, the heating/cooling rate

and the constant temperature time were slowed down to the same as the previous 10 cycles. The specific temperature programs were listed below. The experimental results are presented in Fig. 3e and 3f, and we re-discuss the thermal reliability of MTPCMs based on the thermal cycle results. The figure and revised text are listed below:

Figure:

Fig. 3 Thermal performance of MTPCMs. **a** DSC curves of paraffin and different volume ratio MTPCMs. **b** Comparison of fusion enthalpy, onset melting, and solidifying point of pristine paraffin and MTPCMs

with different volume ratios. **c** TGA curves of paraffin and different volume ratio MTPCMs. **d** Temperature variations of MTPCM during the heat charging/discharging cycles and the magnified heating-freezing process in one period. **e** Over 1000 thermal cycles and **f** four selected typical heat charging/discharging cycles of MTPCM. **g** Thermal conductivities of pristine paraffin and MTPCMs with different volume ratios. **h** Schematic diagram of the directional alignment of NdFeB@Ag particles in paraffin matrix and the consequently anisotropic thermal conductivities of MTPCMs.

Revised text: The experiment system and temperature program are shown in Supplementary Fig. 5 and the Method section, and the MTPCM sample was produced into a 20×20×20 mm cubic block. The cycle was repeated 10 times and the curves are recorded in Fig. 3d.

Thermal reliability is a fatal parameter for PCMs as it can evaluate their service life in practical applications. MTPCMs exhibited almost the same heat charging/discharging rate and phase change time in 10 cycles, presenting their potential for cycle reliability. To verify this point, over 1000 thermal cycles were repeated on a 23.31 % sample (specific temperature setting could be seen in the “Methods” section). The temperature variation versus time of the sample was recorded in Fig. 3e, it could be discovered that there is no obvious fluctuation on the curve. To observe the cycle condition more clearly, we extracted the first, 102nd, 505th, and 1004th four cycles and plotted them in Fig. 3f. The trend of these four curves almost overlaps, proving the very prominent thermal reliability and durability of MTPCMs.

Corresponding temperature programs in the Method section: In a complete heating/freezing process, the sample stayed at 20 °C for 30 min. Then it was heated to 50 °C in 30 min and maintained at this temperature for 30 min. After this stage, the sample was cooled to 20 °C and kept at this temperature point for 30 min. Regarding the over 1000 thermal cycles, we adjusted the heating/cooling rate and the constant temperature time to improve the cycle efficiency. In a complete heating/cooling process, the temperature program was set to stay at 20 °C for 5 min, then it began to heat up to 50 °C in 8 min and stayed at this point for 10 min. In the cooling phase, the temperature cooled to 20 °C in 8 min and stayed at this point for 5 min. Besides, the heating/cooling process was performed once followed the aforementioned charging/discharging program after about every 100 cycles (i.e. stayed at 20 °C for 30 min, heated to 50 °C in 30 min and stayed at this point for 30 min, cooled down to 20 °C in 30 min and stayed at this point for 30 min). Finally, we actually cycled 1054 times.

7. Compressive strength test was not performed properly. What was the targeted compressive strength for the prepared MTPCMs? Also, there was no mention about which International Standards/Code of Practices were followed while conducting this test. What was the compressive strength for the pristine Specimen and how much the deviation that was obtained in the value when compared to the MTPCMs specimens? Please address this carefully and include more discussion of the results, in place of presenting only the results.

Reply: Thank you for your question. The compression tests were done following the China National Standard: GB/T 7314-2017 and we supplement this information in the manuscript. For the compression test, there is no concept of target compressive strength. This test is used to explore the compressive strength of a material. For brittle materials, the compressive strength is the maximum strength during the process of compressing the sample to fragmentation. While for plastic materials, they will appear a yield phenomenon, and the compressive strength is the maximum strength in the compression curves. Based on your advice,

we further conduct the compression test to pure paraffin and add the test result in Fig. 4a to make a comparison between pure paraffin and MTPCMs. We also describe test results of pure paraffin and MTPCMs, discussing the mechanism of why MTPCMs have better compression strength than pure paraffin according to the test result. The revised figure and text are listed below:

Fig. 4 Mechanical performance of MTPCMs. a Compression stress-strain curves of pure paraffin and MTPCMs at the temperature below the melting point. b Compression stress-strain curves of MTPCMs at the temperature above the melting point. The insets are the typical compressed samples. Scale bars: 2mm. c Anisotropic compression stress-strain curves of MTPCM at different directions.

“On this background, the compression tests were conducted on the 10×10×10 mm cubic MTPCM modules and pure paraffin (the measurement system is shown in Supplementary Fig. 7 and the test was carried out following the China National Standard GB/T 7314-2017). We first performed the tests at different temperatures, and the results are depicted in Fig. 4a. At room temperature, compared with pure paraffin, MTPCMs displayed significantly higher compressive yield strength, and this capacity was reinforced about 2.5 to 3.7 times with the increase of volume ratio in the sample. At the temperature above the melting point of paraffin, pure paraffin melted completely into a puddle of liquid, making it impossible to test at all. While to MTPCMs, they could hold their shape until being compressed by an external normal force. At this moment, MTPCMs continuously deformed and were ultimately compressed into a flattened cuboid. No leakage occurred in the whole process, suggesting even though being compressed at a high temperature, MTPCMs can still keep their leakage-proof ability intact (Supplementary Movie 4). These can be observed from the state of typical compressed samples in the insets as well.

It seems to take the result for granted that the compressive strength of MTPCMs is better than pure paraffin. But the mechanism behind this phenomenon is worth discussing. By comparing the two materials, it can easily conclude that the magnetic NdFeB@Ag chains throughout the MTPCMs support the materials giving them a strong structure. They are the basis on which MTPCMs could own excellent compressive yield strength. However, in the high-temperature test, once the paraffin was melted, the magnetic chains yielded rapidly and the compressive yield strength of MTPCMs became almost zero. This indicates that paraffin also plays an important role in the compressive strength of MTPCMs. They tightly surround the NdFeB@Ag chains forming a protective layer that offers a lateral force when the sample is compressed, which helps the magnetic chains keep their original direction and continue to work as supports in the normal direction. Both the paraffin and magnetic chains are indispensable factors and under their combined actions, MTPCMs finally harvest an outstanding compressive strength. Based on the foregoing discussion, it should be also realized that MPTCMs have mechanical anisotropy. Thereby we measured the compressive strength

of MTPCMs in different directions. As predicted, the sample has a better compressive strength in the direction parallel to the magnetic field, originating from the anisotropic structure of MTPCMs. To sum up, compared to conventional PCMs, MTPCMs possess good mechanical performance at room temperature which could broaden their application to the field of green building.”

8. There was no discussion related to the specific heat capacity measurements for MTPCMs, which is highly essential as they play a vital role in phase change kinetics. The thermal properties related findings made on MTPCMs has to get well with the thermal conductivity and specific heat measurements. Else, it becomes a less convincing study. Please address this carefully.

Reply: Thank you for your valuable question. In our experiment, just as the description in the “Characterizations and property measurements” section, the thermal conductivity of MTPCMs was calculated by the equation $\lambda = \alpha \times C_p \times \rho$, where α represents the thermal diffusivity, C_p is specific heat capacity, and ρ is density. For phase change materials, we usually put more emphasis on their fusion enthalpy and thermal conductivity as these two parameters are related to the heat conversion and storage and heat transfer ability of PCMs. Hence, we put the thermal conductivity, the onset melting/solidifying point, and fusion enthalpy of MTPCMs in the manuscript. As for the specific heat capacity, we use it to calculate the thermal conductivity. In our work, this parameter was tested by the DSC instrument and the engineer in NETZSCH told us that among the existing equipment for testing the specific heat capacity, the DSC instrument has the highest accuracy which is 95~97 %. So in general, the specific heat capacity curve needs not be tested many more times. For the three different volume ratios MTPCMs, their specific heat capacities at 30 °C are listed below:

MTPCM Samples	15.84%	19.12%	23.31%
$C_p/J/cm^3 \cdot K$	4.14	3.68	3.51

9. For the developed form-stable MTPCMs, it would be wise to report their encapsulation ratio, encapsulation efficiency and thermal storage capability. Without these parameters, it becomes very difficult to substantiate the usefulness of the prepared PCMs.

Reply: Thank you for your kind reminding. We are sorry that we didn’t explain these issues clearly in the manuscript. As for the MTPCMs, hard magnetic particles--NdFeB@Ag particles perform the encapsulation function and paraffin are the real phase change materials. Hence, it is valuable to explore the lowest mixing ratio for MTPCMs and we only mentioned the lowest volume ratio in the manuscript. According to your advice, we have revised the text in the manuscript as follows:

“It was explored that the lowest volume mixing ratio for MTPCMs was 15.84 %, and as a comparison, we prepared three samples with different volume ratios of 15.84 %, 19.12 %, and 23.31 % in this paper.”

As for the encapsulation efficiency, we have described the method in detail in the “Surface modification of NdFeB” and “Preparation of MTPCMs” sections. For the surface modification of NdFeB, our method are efficient and the weight of each reagent involved in the reaction can be scaled up in proportion, and consequently, the production of NdFeB@Ag can be greatly increased. For the preparation of MTPCMs, the magnetization process only costs 30-60 s and the time is mainly spent on heating and cooling MTPCMs. However, the efficiency could be enhanced by customizing a larger mold and employing more efficient

heating and cooling equipment. Based on your advice, we added several sentences to supplement this information in the manuscript.

“The weight of each reagent involved in the reaction can be scaled up in proportion. Consequently, the production of NdFeB@Ag can be increased.”

“It could be concluded that the magnetization process cost a very short time, and the time was mainly spent on heating and cooling the materials. Therefore, the efficiency of fabricating MTPCMs could be improved by customizing a larger mold and employing more efficient heating and cooling equipment.”

To phase change materials, their thermal storage capacity is reflected on their latent heat or fusion enthalpy. In the manuscript, we did the DSC test for three different volume ratios MTPCMs and the fusion enthalpy can be obtained from the DSC curves. We mentioned that “The fusion enthalpy for neat paraffin was 163.75 J cm^{-3} , as the addition of NdFeB@Ag particles, the value didn't decrease but instead changed to 178.70 , 172.59 and 141.63 J cm^{-3} , which was ascribed to the magnetic particles increased the density of MTPCMs and thus improved their energy storage density”. Besides, the thermal storage capacity of MTPCMs can also reflect by the heat-to-electricity conversion and storage experiment, in which the MTPCM converse its latent heat to 4.47J electrical energy. In the thermal management experiment, the MTPCM could control the temperature of the object for a long time, which can also demonstrate its superior thermal storage capacity. And we have emphasized this point in the manuscript, such as “These data suggest the large latent heat and superior heat-to-electricity conversion and storage performance of MTPCMs.”, “The aforesaid two experiments prove that MTPCMs have superb thermal management capacity”, and “It can be calculated that MTPCMs reduce the battery temperature rise by 31.39% and this is attributed to the high thermal conductivity and large latent heat of MTPCMs.”

10. Conclusion must be well supported with the data.

Reply: Thank you for your valuable comments. Combined with the comments of reviewer 3 that the summary of the article does not adequately reflect the work done. We have revised our discussion section, corresponding data are added followed by the conclusion. We also supplement more conclusions with data to reflect the work done elaborately. Thank you again for your kind advice and we hope it will improve the quality of our manuscript. The revised text is listed as follows:

“The surface magnetic flux density of the 15.85% MTPCM reaches over 30 mT , and the magnetism of MTPCMs can maintain under a high temperature or over long periods.”

“In the thermal aspect, the NdFeB@Ag particles just function as the supporting material and have no impact on paraffin. The values of onset melting and solidifying point for pure paraffin (38.65 and $36.23 \text{ }^\circ\text{C}$) and the three different ratios MTPCMs (38.95 and $37.33 \text{ }^\circ\text{C}$ for 15.84% , 39.5 and $37.13 \text{ }^\circ\text{C}$ for 19.12% and 40.05 and $37.07 \text{ }^\circ\text{C}$ for 23.31% sample) are nearly the same. As for the fusion enthalpy, owing to the synergistic effect of the mixing ratio and density increase in MTPCMs, this parameter turns from 163.75 J cm^{-3} for paraffin to 178.70 , 172.59 , and 141.63 J cm^{-3} for the three MTPCMs. Thereby, it could be concluded that MTPCMs are not affected by the magnetic particles and still retain the proper melting/solidifying point and large latent heat. Except that, the TGA test presents that the addition of

NdFeB@Ag particles delays the 5 % weight loss temperature of MTPCMs by 30 to 65 °C, making MTPCMs possess excellent thermostability. As for the heat charging/discharging cycles, MTPCMs exhibit an efficient thermal storage/release rate. Combined with the result of 1000 thermal cycles, MTPCMs demonstrate their superior thermal reliability and application prospect.”

“Moreover, the interconnected structure provides heat transfer pathways, combined with the superior thermal conduction of NdFeB@Ag together enhancing the thermal conductivity of MTPCMs by 14~16 times than pristine paraffin, reaching 2.97, 3.11, and 3.41 W m⁻¹ K⁻¹ for the three different volume ratio MTPCMs.”

“Compared with the compressive yield strength of paraffin which is only 2.1 MPa, this value increases to at least 2.5 times reaching 5.19 MPa for MTPCMs at room temperature.”

“, consequently endowing MTPCMs with the electrical conductivity of the order of 10⁴ S m⁻¹. As a result, the electricity-to-heat harvest efficiency of MTPCM reaches as high as 78.45 %, and a small 20 × 20 × 10 mm MTPCM block can harvest a 4.47 J net electrical energy from its latent heat. These illustrate the high energy conversion efficiency and energy harvesting ability of MTPCMs in electricity and heat conversion.”

“When the battery discharges at a current of 17 A, the MTPCM function its heat dissipation role reducing the temperature rise of the battery by 31.39 %. While at low temperature, the MTPCM protects the battery by converting electricity to heat, regulating its temperature from -20 °C to keep it above 0 °C.”

Reviewer #3:

This paper presents "Magnetically tightened form-stable phase change materials with modular assembly and geometric conformality features". I am convinced that such preparing magnetically tightened form-stable phase change materials in this article have a guiding significance for the preparation of PCM composites. The authors did a solid job. While the article has significant shortfalls as presented below. In my opinion, the following comments must be addressed before considering the publication.

Reply: Thank you so much for your kind helps with our manuscript. We deeply appreciate your time and patience in the review process. Your valuable comments have been of enormous help for us to improve our work. Based on your kind comments and suggestions, we have made corresponding changes in our revised manuscript. Our answers to your specific questions are listed below and changes that have been made to the manuscript are highlighted in red for your review convenience. Again, thank you very much for your help with the improvements of our manuscript and we hope this revision can get your approval.

1. The summary of the article does not adequately reflect the work done.

Reply: Thank you for your professional suggestions. We agree with your suggestion and based on your advice, we have revised and supplemented more conclusions and data in the discussion section to better reflect the work we have done. We sincerely appreciate your suggestion which indeed helps us improve the clarity and quality of our manuscript. The revised section is listed as follows:

“In the thermal aspect, the NdFeB@Ag particles just function as the supporting material and have no impact on paraffin. The values of onset melting and solidifying point for pure paraffin (38.65 and 36.23 °C) and the three different ratios MTPCMs (38.95 and 37.33 °C for 15.84 %, 39.5 and 37.13 °C for 19.12 % and 40.05 and 37.07 °C for 23.31% sample) are nearly the same. As for the fusion enthalpy, owing to the synergistic effect of the mixing ratio and density increase in MTPCMs, this parameter turns from 163.75 J cm⁻³ for paraffin to 178.70, 172.59, and 141.63 J cm⁻³ for the three MTPCMs. Thereby, it could be concluded that MTPCMs are not affected by the magnetic particles and still retain the proper melting/solidifying point and large latent heat. Except that, the TGA test presents that the addition of NdFeB@Ag particles delays the 5 % weight loss temperature of MTPCMs by 30 to 65 °C, making MTPCMs possess excellent thermostability. As for the heat charging/discharging cycles, MTPCMs exhibit an efficient thermal storage/release rate. After over 1000 thermal cycles, the heat charging/discharging curve of MTPCM almost overlaps with the first cycle curve, demonstrating their superior thermal reliability and long service life in practical applications. Moreover,”

“Compared with the compressive yield strength of paraffin which is only 2.1 MPa, this value increases to at least 2.5 times reaching 5.19 MPa for MTPCMs at room temperature.”

“consequently endowing MTPCMs with the electrical conductivity of the order of 10⁴ S m⁻¹. As a result, the electricity-to-heat harvest efficiency of MTPCM reaches as high as 78.45 %, and a small 20×20×10 mm MTPCM block can harvest a 4.47 J net electrical energy from its latent heat. These illustrate the high energy conversion efficiency and energy harvesting ability of MTPCMs in electricity and heat conversion.”

“When the battery discharges at a current of 17 A, the MTPCM function its heat dissipation role reducing the temperature rise of the battery by 31.39 %. While at low temperature, the MTPCM protects the battery by converting electricity to heat, regulating its temperature from -20 °C to keep it above 0 °C.”

2. 10 thermal cycles are not enough for new materials. Authors are required to include their thermal and chemical analysis after at least 200 thermal cycles.

Reply: Thank you for your professional comments and we agree with your suggestion for at least 200 thermal cycles. We have done 1054 thermal cycles to verify the thermal reliability and durability of MTPCMs. We put the data of 1054 thermal cycles and the extracted typical four cycle curves in Fig. 3e and 3f and discussed the test results in the manuscript. Besides, the specific temperature program was put in the Method section. The text and figure are shown as follows::

Text: “Thermal reliability is a fatal parameter for PCMs as it can evaluate their service life in practical applications. MTPCMs exhibited almost the same heat charging/discharging rate and phase change time in above 10 cycles, presenting their potential for cycle reliability. To verify this point, over 1000 thermal cycles were repeated on a 23.31 % sample (specific temperature setting could be seen in the “Methods” section). The temperature variation versus time of the sample was recorded in Fig. 3e, it could be discovered that there is no obvious fluctuation on the curve. To observe the cycle condition more clearly, we extracted the first, 102nd, 505th, and 1004th four cycles and plotted them in Fig. 3f. The trend of these four curves almost overlaps, proving the very prominent thermal reliability and durability of MTPCMs.”

Fig.3:

Fig. 3 Thermal performance of MTPCMs. a DSC curves of paraffin and different volume ratio MTPCMs. b Comparison of fusion enthalpy, onset melting, and solidifying point of pristine paraffin and MTPCMs with different volume ratios. c TGA curves of paraffin and different volume ratio MTPCMs. d Temperature variations of MTPCM during the heat charging/discharging cycles and the magnified heating-freezing process in one period. e Over 1000 thermal cycles and f four selected typical heat charging/discharging cycles of MTPCM. g Thermal conductivities of pristine paraffin and MTPCMs with different volume ratios. h

h Schematic diagram of the directional alignment of NdFeB@Ag particles in paraffin matrix and the consequently anisotropic thermal conductivities of MTPCMs.

Method section: In a complete heating/freezing process, the sample stayed at 20 °C for 30 min. Then it was heated to 50 °C in 30 min and maintained at this temperature for 30 min. After this stage, the sample was cooled to 20 °C and kept at this temperature point for 30 min. Regarding the over 1000 thermal cycles, we adjusted the heating/cooling rate and the constant temperature time to improve the cycle efficiency. In a complete heating/cooling process, the temperature program was set to stay at 20 °C for 5 min, then it began to heat up to 50 °C in 8 min and stayed at this point for 10 min. In the cooling phase, the temperature cooled to 20 °C in 8 min and stayed at this point for 5 min. Besides, the heating/cooling process was performed once followed the aforementioned charging/discharging program after about every 100 cycles (i.e. stayed at 20 °C for 30 min, heated to 50 °C in 30 min and stayed at this point for 30 min, cooled down to 20 °C in 30 min and stayed at this point for 30 min). Finally, we actually cycled 1054 times.

3. Leakage tests are one of the most important analyses tests for the PCM composites. Could you please give information about the leakage test time applied in this study? Even if the PCM composite leakage, it is difficult to distinguish with the naked eye in short times. Specifically, the prepared samples need to be heated to 50 °C on the filter paper for approximately 2 hours to observe whether there are traces of liquid leakage on the filter paper during the phase transition.

Also, not only a digital image but also DSC, SEM, FTIR, and TEM image is necessary to demonstrate if the PCM leaked during the leakage test.

Reply: Thank you for your question. In the manuscript, we conducted several tests to directly demonstrate the leakage-proof property of MTPCMs and the results are in Fig. 1d and Supplementary Movie 1. Besides, the leakage-proof feature of MTPCMs can also be illustrated indirectly in Supplementary Movie 2, Movie 4, Fig 6b, and Movie 7. In Fig. 1d and Movie 1, pure paraffin, unmagnetized composites, and MTPCM with the critical volume ratio of 15.84% were heated on the heating platform or by a heat gun. As we can directly observe the melting of the three different samples, we did not record the time. However, the MTPCM sample has better thermal conductivity than pure paraffin and the unmagnetized composite, and the three samples are heated under the same heating situation. It can be confirmed that if the pure paraffin and unmagnetized composite are melting, the paraffin in the MTPCM sample is also melting. So it is reasonable to conclude the leakage-proof ability of MTPCMs by the experiment.

In addition, in Supplementary Movie 2, it can be seen by naked eyes that the MTPCM sample is completely melted because it can be kneaded to any shape. And no leakage occurred during this process which can demonstrate the leakage-proof of MTPCMs.

As for Fig. 6b and Movie 7, under the same heating condition, pure paraffin begins to melt at the 25th second and drops to the table after being heated for 6 minutes. Considering the MTPCM has better thermal conductivity than pure paraffin, it can be sure that the paraffin in MTPCM is also melted from the 25th second. But the MTPCM doesn't leak in the whole process which can be directly observed that there is no paraffin droplet falling to the table. The above experiments all prove the superior leakage-proof ability of MTPCMs.

We also accept your advice and add two experiments to illustrate the leakage-proof feature of MTPCMs. First, we take MTPCM with the lowest volume ratio--15.84% as a typical sample and heat this sample in the oven at 50 °C for over 2h. Except for taking photos of the filter paper before and after the heating

process, we also weight the sample by an analytical balance. The results are put in Supplementary Fig.3, the Fig and the corresponding revised text are shown as follows. Besides, it is necessary to point out that this method is the most commonly used method to verify the leakage-proof ability of PCM materials.

Supplementary Fig.3:

Supplementary Figure 3. Leakage-proof result of MTPCMs. a Leakage test setup and the morphology of the MTPCM sample before and after heating. b The weight of the MTPCM sample before and after heating.

The revised text: Furthermore, we took the 15.84% MTPCM as a sample and heated it at 50 °C for over two hours in the oven. It can be seen that the weight of this sample did not change at all before and after heating, demonstrating the excellent leakage-proof ability of MTPCMs again (see Supplementary Fig.3).

Additionally, we also do the DSC test for the MTPCM sample before and after being heated at 50 °C for over 2h. It can be seen from the DSC result that the fusion enthalpy of MTPCM before and after being heated are nearly the same, demonstrating the leakage-proof ability of MTPCM.

From the DSC test, it can be concluded that the 15.84% MTPCM sample has similar fusion enthalpy, which proves the leakage-proof feature of MTPCM again.

4. Only EDX spectra cannot prove “the interactions of magnetic material and PCM during the preparation of form stable composites are physical and no chemical interaction occurs.”. More measurements such as Raman, FTIR are needed to certify this statement.

Reply: Thank you for your question. In our manuscript, the EDX result is mainly used to illustrate the anisotropic structure of MTPCMs. We did the XRD test to detect whether there is any new substance formed between pure paraffin and NdFeB@Ag particles. The XRD result shows that NdFeB@Ag particles and paraffin are a physical combination and no new substance is formed. According to your advice, we supplement an FTIP test to further confirm the above conclusion, and the test result is listed below:

We test the FTIR spectra of the 15.84% MTPCM sample and pure paraffin to make a comparison. It can be seen from the result that MTPCM has almost the same spectra as pure paraffin. There aren't any new spectra emerge in the FTIR spectra of MTPCM. This result demonstrates that paraffin and NdFeB@Ag particles are a physical combination and no chemical reaction occurs.

REVIEWERS' COMMENTS

Reviewer #2 (Remarks to the Author):

The authors have thoroughly revised their manuscript and incorporated the required changes as per the comments given. The revised manuscript is now acceptable for publication.

Reviewer #3 (Remarks to the Author):

This is good work. Thank you for your responses. It is accepted

Response to Editors and Reviewers

Point-by-point reply

Reviewer #2:

The authors have thoroughly revised their manuscript and incorporated the required changes as per the comments given. The revised manuscript is now acceptable for publication.

Reply: The authors are glad to see that our response satisfactorily addressed the comments. Also, the authors sincerely thank the reviewer for supporting our manuscript for publication.

Reviewer #3:

This is good work. Thank you for your responses. It is accepted.

Reply: The authors really appreciate this comment. Additionally, the authors would like to thank the reviewer for supporting the current version of the manuscript for publication.